# Actin-microtubule cytoskeletal interplay mediated by MRTF-A/SRF signaling promotes dilated cardiomyopathy caused by *LMNA* mutations

Caroline Le Dour [1,13], Maria Chatzifrangkeskou[1,13], Coline Macquart[1,13], Maria M. Magiera [2,3], Cécile Peccate[1], Charlène Jouve [4], Laura Virtanen[5], Tiina Heliö[6], Katriina Aalto-Setälä [7], Silvia Crasto[8,9], Bruno Cadot[1], Déborah Cardoso[1], Nathalie Mougenot[10], Daniel Adesse [11], Elisa Di Pasquale [8,9], Jean-Sébastien Hulot[4], Pekka Taimen [5,12], Carsten Janke [2,3] & Antoine Muchir [1] ✉

Mutations in the lamin A/C gene (*LMNA*) cause dilated cardiomyopathy associated with increased activity of ERK1/2 in the heart. We recently showed that ERK1/2 phosphorylates cofilin-1 on threonine 25 (phospho(T25)-cofilin-1) that in turn disassembles the actin cytoskeleton. Here, we show that in muscle cells carrying a cardiomyopathy-causing *LMNA* mutation, phospho(T25)-cofilin-1 binds to myocardin-related transcription factor A (MRTF-A) in the cytoplasm, thus preventing the stimulation of serum response factor (SRF) in the nucleus. Inhibiting the MRTF-A/SRF axis leads to decreased α-tubulin acetylation by reducing the expression of *ATAT1* gene encoding α-tubulin acetyltransferase 1. Hence, tubulin acetylation is decreased in cardiomyocytes derived from male patients with *LMNA* mutations and in heart and isolated cardiomyocytes from *Lmna*[p.H222P/H222P] male mice. In *Atat1* knockout mice, deficient for acetylated α-tubulin, we observe left ventricular dilation and mislocalization of Connexin 43 (Cx43) in heart. Increasing α-tubulin acetylation levels in *Lmna*[p.H222P/H222P] mice with tubastatin A treatment restores the proper localization of Cx43 and improves cardiac function. In summary, we show for the first time an actin-microtubule cytoskeletal interplay mediated by cofilin-1 and MRTF-A/SRF, promoting the dilated cardiomyopathy caused by *LMNA* mutations. Our findings suggest that modulating α-tubulin acetylation levels is a feasible strategy for improving cardiac function.

Dominant mutations in *LMNA*, which encodes nuclear lamin A/C, can cause dilated cardiomyopathy with conduction system disease[1]. Patients with *LMNA* mutations are characterized by a poor prognosis compared with non-carriers, with a high rate of major cardiac events such as high degree atrioventricular block, sudden death due to malignant ventricular tachycardia, and end-stage heart failure[2]. Currently, no drugs are curative, and heart transplantation is frequently necessary for these patients.

ERK1/2 is hyperactivated in in vitro and in vivo models with *LMNA* mutations, and recent findings have provided proof of principle for ERK1/2 inhibition as a therapeutic option to prevent or delay the onset of heart failure in patients with dilated cardiomyopathy caused by mutations in *LMNA*[3–6]. However, ERK1/2 signaling is a central signaling pathway that regulates a wide variety of cellular processes. Downstream effectors remaining largely unknown, we aim to decipher molecular mechanisms involved in order to raise more specific therapeutic options.

The development of dilated cardiomyopathy caused by mutations in *LMNA* is associated with both actin[7,8] and microtubule[9,10] network alterations. Actin–microtubule crosstalk is particularly important for the regulation of cell mechanical resistance[11]. Impairment in this latter leads to the pathogenesis and progression of dilated cardiomyopathy, and a better understanding of this mechanism could lead to the design of precision therapeutics[12].

In cardiomyocytes, the actin cytoskeleton acts as a scaffold, regulating cell shape, providing mechanical integrity and resistance, and stabilizing contractile units called sarcomeres. This structural framework mediates biomechanical signaling, altering gene expression and posttranslational processing in response to strain, and directly participates in the remodeling of cardiac muscle[13,14]. Dynamic rearrangement of the actin cytoskeleton is achieved via the controlled nucleation and elongation of filaments (F-actin) from monomers (G-actin) and their disassembly; these processes are tightly regulated in time and space by numerous actin-binding proteins, scaffolding proteins, and upstream signaling factors. Cofilin-1 is an essential actin cytoskeleton-regulating protein[15]. The activity and subcellular localization of cofilin-1 depend on its phosphorylation state. Cofilin-1 phosphorylated at serine 3 (S3) is inactive and remains in the cytosol, while the non-phosphorylated (S3) form is active and able to severe and depolymerize F-actin[16]. We recently showed that cofilin-1 phosphorylated at threonine 25 (T25) by ERK1/2 becomes active and disassembles actin filaments, including sarcomeric F-actin, in *Lmna*[p.H222P/H222P] mice with dilated cardiomyopathy[7]. Actin dynamics regulate the myocardin-related transcription factor A (MRTF-A)/serum response factor (SRF) axis to enable transcription of a subset of genes encoding myogenic contractile and cytoskeletal proteins[17–22]. Inhibition of the MRTF-A/SRF axis promotes the development of cardiomyopathy and heart failure[8,23,24].

Meanwhile, the microtubules in cardiac myocytes align predominantly along the longitudinal axis of the cell and interact with sarcomeres[25]. The microtubule cytoskeleton is well known for its role as a track for cargo transport, and it also functions as a mechanotransducer, converting contraction forces into cellular signals in cardiac myocytes[26,27]. Microtubules assemble from highly conserved α/β-tubulin heterodimers and can undergo a wide variety of posttranslational modifications (PTMs) that regulate their physiological functions[28–31]. Recent studies revealed that excessive detyrosination PTM of heart microtubules affects mechanotransduction and is associated with cardiomyopathy and heart failure[32–34]. This suggests that the regulation of the microtubule cytoskeleton by tubulin PTMs plays a key role in cardiac myocyte function. However, very little is known about the potential roles of other PTMs in this process.

Here, we aimed to elucidate the mechanisms underlying the interplay and possible synergy between these two cytoskeletal components in cardiomyopathy caused by mutations in *LMNA*. We show that cofilin-1 phosphorylated on T25 binds to MRTF-A in the cytoplasm in muscle cells expressing *LMNA* mutations. We find that inhibition of the MRTF-A/SRF actin-dependent transcriptional circuit regulates α-tubulin acetylation by decreasing the expression of *ATAT1*, which encodes α-tubulin acetyltransferase 1. We next discover that a reduction in α-tubulin acetylation in mice leads to a mislocalization of gap junction protein Cx43 in cardiomyocytes, disturbing cardiac contractility and resulting in heart malfunction. Finally, we are able to restore proper Cx43 localization at intercalated discs, and thereby improve cardiac function, by increasing α-tubulin acetylation via tubastatin A administration to inhibit the α-tubulin deacetylase HDAC6. As α-tubulin deacetylation through HDAC6 activity has already been established as a possible therapy in cancer clinical trials[35], we suggest that its targeting may be a viable therapeutic option for cardiomyopathy caused by mutations in *LMNA*.

## Results

### MRTF-A subcellular localization is altered in cellular and mouse models expressing cardiomyopathy-causing mutant A-type lamins

We previously highlighted that elevated ERK1/2 signaling catalyzes cofilin-1 phosphorylation on T25 and alters actin dynamics in dilated cardiomyopathy caused by mutations in *LMNA*[7]. Given that actin dynamics regulate the MRTF-A/SRF axis[17,18,20–22], we wondered whether the latter is altered in the presence of mutated A-type lamins. To assess the roles of elevated ERK1/2 activity and actin defects in *LMNA* cardiomyopathy, we utilized C2C12 cells, a simple and well-characterized model system that is frequently used as a model and can adopt some features of cardiomyocytes[36,37]. Immunoblotting of fractionated cells indicated that the expression of MRTF-A was significantly increased in the cytoplasmic fraction relative to the nuclear fraction in C2C12 cells expressing mutant lamin A (C2-H222P) compared with C2C12 cells expressing wild-type lamin A (C2-WT) (Fig. 1a). Consistently, increased cytoplasmic MRTF-A localization in C2-H222P cells was detected by immunofluorescence microscopy (Fig. 1b). These results are in agreement with data from a previous work[8]. The abnormal cellular localization of MRTF-A in C2-H222P cells was further blunted by treatment with selumetinib, which inhibits MEK1/2, the kinases that specifically phosphorylate ERK1/2 (Fig. 1b, c). In cardiomyocytes derived from patient's induced pluripotent stem cells (iPS-CMs) carrying *LMNA* mutations, expression of phosphorylated-ERK was elevated (Fig. 1d), confirming abnormal ERK1/2 signaling activation, and cytoplasmic localization of MRTF-A was observed in immunofluorescence (Fig. 1e). Similar results were found in cardiac sections from *Lmna*[p.H222P/H222P] mice, a model of dilated cardiomyopathy caused by *LMNA* mutations[38] that has been shown to be relevant to understanding the pathogenesis of heart disease[7,39–41] and to test the effect of novel therapeutic compounds[3,42]. Immunostaining of MRTF-A shows that nuclear staining is decreased as compared with wild-type mice (Fig. 1f). Altogether, these results suggest that the cytoplasmic localization of MRTF-A is driven by abnormal ERK1/2 signaling in cardiomyopathy caused by *LMNA* mutations.

### Phospho(T25)-cofilin-1 binds to MRTF-A and alters its subcellular localization in cells expressing cardiomyopathy-causing mutant A-type lamins

As elevated ERK1/2 signaling alters actin dynamics by catalyzing cofilin-1 phosphorylation on T25, we next aimed to investigate the role of cofilin-1 phosphorylation in MRTF-A subcellular localization. Although we show overexpression of total cofilin-1 in C2-H222P cells (Fig. 2a) (previously described as resulting from impaired degradation by the proteasome[43]), expression of cofilin-1 phosphorylated on S3 was unchanged, suggesting its relative decreased phosphorylation. In the meantime, we confirmed that the relative phosphorylation of cofilin-1 on T25 was significantly increased in C2-H222P cells, as previously reported[7] (Fig. 2a). Further, while the cellular localization of cofilin-1 was similar in C2-H222P and C2-WT cells, phospho(T25)-cofilin-1 was mostly localized in the cytoplasm in C2-H222P cells, whereas it was primarily nuclear in C2-WT cells (Fig. 2b). We, therefore, hypothesized that increased phospho(T25)-cofilin-1 expression alters the subcellular localization of MRTF-A. To test this hypothesis, we ectopically expressed cofilin-1(T25A), a nonphosphorylatable variant, and cofilin-1(T25D), a phosphomimetic variant in C2-H222P cells. Immunoblotting

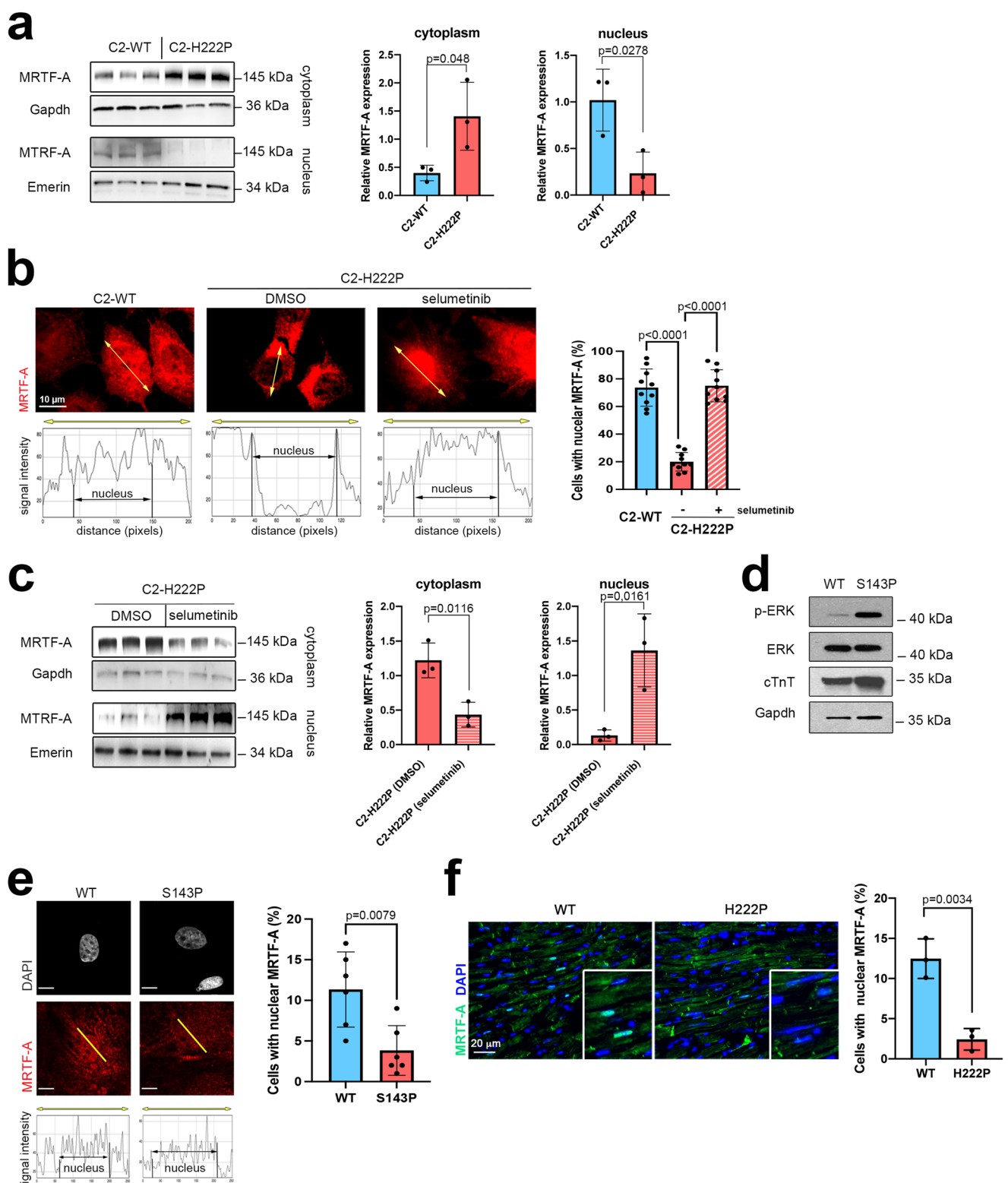

of extracts from C2-H222P cells transfected with the nonphosphorylatable cofilin-1(T25A) variant show an increased F/G actin ratio unlike nontransfected cells and cells transfected with phosphomimetic cofilin-1(T25D), confirming phospho(T25)-cofilin-1 alters actin dynamics (Fig. 2c). Immunostaining of MRTF-A shows predominantly nuclear localization in C2-H222P cells transfected with the nonphosphorylatable cofilin-1(T25A) variant while nontransfected cells and cells transfected with phosphomimetic cofilin-1(T25D) show mostly cytoplasmic MRTF-A localization (Fig. 2d). We next transfected

C2-H222P cells with a nuclear export consensus sequence (NES) cofilin-1 mutant, cofilin-1(V20A), to prevent the active translocation of cofilin-1 from the nucleus to the cytoplasm. In these cells, the F/G actin ratio did not differ from that in nontransfected cells and MRTF-A was restricted to the nucleus (Fig. 2c, d). These results suggest that cofilin-1 phosphorylation on T25 plays a key role in the cellular localization of MRTF-A in muscle cells.

To further confirm that phosphorylation of cofilin-1 on T25 influences the cellular distribution of MRTF-A, we sought to investigate the

**Fig. 1 | Abnormal MRTF-A localization in cells expressing mutated A-type lamins. a** Immunoblots showing the MRTF-A expression in nuclear and cytoplasmic extracts from C2-WT and C2-H222P cells (*n* = 3 independent experiments). Emerin and GAPDH were used as loading controls for the nuclear and cytoplasmic fractions, respectively. Bar graph shows the quantification of MRTF-A (*n* = 3 independent experiments, mean ± SD). **b** Representative immunofluorescence micrographs of MRTF-A staining in C2-WT and C2-H222P cells treated or not with selumetinib to inhibit the phosphorylation of ERK1/2. Scan line graphs represent the intensity of MRTF-A staining along the yellow arrow lines. Bar graph shows the quantification of nuclear MRTF-A (*n* = 250, mean ± SD). Statistics: one-way ANOVA followed by Tukey's multiple comparison test. **c** Immunoblots showing the MRTF-A expression in nuclear and cytoplasmic extracts of C2-H222P cells treated or not with selumetinib (*n* = 3 independent experiments). Emerin and GAPDH were used as loading controls for the nuclear and cytoplasmic fractions, respectively. Bar graph shows the quantification of MRTF-A (*n* = 3 independent experiments,

mean ± SD). **d** Immunoblots showing the p-ERK1/2, ERK1/2, cTnT, and GAPDH expression in protein extracts from cardiomyocytes derived from patient-specific human iPSCs carrying a *LMNA* mutation (p.S143P) and control (WT). **e** Representative micrographs showing the MRTF-A labeling of cardiomyocytes derived from control (WT) and patient-specific human iPSCs carrying the *LMNA* p.S143P mutation (S143P). Scan line graphs represent the intensity of MRTF-A staining along the yellow arrow lines. Scale bar 10 μm. The bar graph shows the quantification of nuclear MRTF-A (*n* = 350 cells, mean ± SD). **f** Fluorescence micrographs showing the MRTF-A labeling of heart cross-sections from 3-month-old male wild-type (WT) and *Lmna*[p.H222P/H222P] (H222P) mice. Nuclei counterstained with DAPI are also shown. Bar graph shows the quantification of cardiomyocytes with nuclear MRTF-A staining from heart sections of three different mice (*n* = 150 cells, mean ± SD). Statistics: for **a**, **c**, **e**, and **f**, unpaired two-tailed *t*-test. Source data are provided as a Source Data file.

effects of cofilin-1 dephosphorylation. Three phosphatases might dephosphorylate cofilin-1 on T25: pyridoxal phosphatase (PDXP), protein phosphatase slingshot homolog 1 (SSH1), and serine/threonine-protein phosphatase 2B (PP2B)[44]. We transiently transfected C2-WT cells with siRNAs targeting *PDXP*, *SSH1*, and *PP2B* mRNA and examined cofilin-1 phosphorylation and MRTF-A localization in these cells via immunoblots and immunofluorescence staining, respectively. The siRNA-induced depletion of PDXP in C2-WT cells led to increased phospho(T25)-cofilin-1 expression (Fig. 2e) and consequent restriction of MRTF-A to the cytoplasm (Fig. 2f). We concluded that the phosphorylation of cofilin-1 on T25 drives the localization of MRTF-A.

We next investigated whether phospho(T25)-cofilin-1 could alter the cellular distribution of MRTF-A by direct binding. To test this hypothesis, we first transfected C2-H222P cells with Flag-tagged MRTF-A, performed immunoprecipitations (IPs) using antibodies against Flag or MRTF-A, and immunoblotted with antibody against phospho(T25)-cofilin-1. We reported an interaction between MRTF-A and phospho(T25)-cofilin-1 (Fig. 2g top). Further, we found that MRTF-A interacts only with cofilin-1 and phospho(T25)-cofilin-1, but not phospho(S3)-cofilin-1, which is known to regulate cofilin-1 function[45] (Fig. 2g bottom). We then used an in situ proximity ligation assay and showed enhanced interactions between phospho(T25)-cofilin-1 and MRTF-A in C2-H222P cells compared with C2-WT cells (Fig. 2h). Finally, using extracts from C2-WT and C2-H222P cells and an antibody against MRTF-A, we showed that phospho(T25)-cofilin-1 coimmunoprecipitated with MRTF-A and that this interaction was more pronounced in C2-H222P cells (Fig. 2i). Altogether, these results show that phospho(T25)-cofilin-1 directly interacts with MRTF-A and that this interaction influence the cellular localization of MRTF-A.

## SRF activity is decreased in cells expressing cardiomyopathy-causing mutant A-type lamins

MRTF-A acts as a transcriptional coactivator to stimulate SRF-dependent gene expression. Therefore, we wondered whether MRTF-A altered localization due to increased phospho(T25)-cofilin-1 expression might cause impaired SRF activity in *LMNA*-induced cardiomyopathy. To evaluate this hypothesis, we first assessed SRF promoter activity with a luciferase assay (schematically shown in Fig. 3a), revealing decreased SRF promoter activity in C2-H222P cells compared with C2-WT cells (Fig. 3b). Moreover, selumetinib treatment (inhibiting ERK1/2 activity) of C2-H222P cells restored the SRF activity to levels compared to that in untreated C2-WT cells (Fig. 3b). These results suggested a correlation between the expression of mutant Lamin A, ERK1/2 signaling and SRF activity.

We next assessed whether the modulation of cofilin-1 expression and T25 phosphorylation could alter SRF activity. The overexpression of a plasmid encoding WT cofilin-1 led to significantly decreased SRF activity compared with that in nontransfected cells in both C2-H222P and C2-WT cells (Fig. 3b). Additionally, transfection with plasmids

expressing nonphosphorylatable cofilin-1(T25A) had no effect on SRF activity, while transfection with plasmids expressing phosphomimetic cofilin-1(T25D) led to decreased SRF activity compared with that in nontransfected cells (Fig. 3b). Finally, in C2-H222P cells transfected with a plasmid encoding cofilin-1 with a mutated NES (V20A), the SRF activity was significantly increased compared to that in nontransfected cells (Fig. 3b). These results suggest that phosphorylation of cofilin-1 on T25 is correlated to SRF repression via modulation of MRTF-A localization.

To further evaluate SRF transcriptional activity, we used RT-qPCR to assess the mRNA expression of several known SRF transcriptional targets[46], including α2-actin (*Acta2*), four-and-a-half-LIM-only 2 (*Fhl2*), troponin C1 (*Tnnc1*), myomesin 1 (*Myom1*), supervillin (*Svil*) and moesin (*Msn*) in C2-WT and C2-H222P cells. As expected, all the above-mentioned transcripts were significantly downregulated in C2-H222P cells (Fig. 3c). Similarly, the mRNA levels of *Srf*, *Acta2*, and *Fhl2* were downregulated in cardiac tissue from *Lmna*[p.H222P/H222P] mice compared with control hearts at 6 months of age when cardiac function is altered[47] (Fig. 3d), suggesting downregulation of SRF-driven transcription along the progression of the disease.

## Phospho(T25)-cofilin-1 impedes SRF regulation and leads to cardiomyopathy in mice

We next assessed the effect of cofilin-1 on SRF activity in vivo. We injected adeno-associated virus (AAV) vectors expressing WT cofilin-1 as well as AAV vectors expressing the mutant nonphosphorylatable cofilin-1(T25A) into 3-month-old WT mice (schematically shown in Fig. 3e). Three months following injection, the mice showed increased cofilin-1 expression in the heart (Fig. 3f, top panel). In hearts from *Lmna* WT mice overexpressing WT cofilin-1, as well as in hearts from *Lmna*[p.H222P/H222P] mice, the actin dynamics (F/G ratio) was altered with predominant globular actin (Fig. 3f, bottom panel), and cardiac function was significantly decreased as determined by ejection fraction (Table S1) and fractional shortening values (Fig. 3g). This was not observed in hearts from untransduced *Lmna* WT mice (UT) or over-expressing nonphosphorylatable cofilin-1(T25A). We next compared cardiac tissue gene expression in control WT mice, WT mice transduced with an AAV vector expressing either WT cofilin-1 or cofilin-1(T25A) and *Lmna*[p.H222P/H222P] mice using genome-wide microarray analysis GSE218891. All of the groups showed clear separation according to principal component analysis (Fig. 3h) and a heat map constructed by unsupervised hierarchical clustering analysis (Fig. 3i). We next used a supervised learning method to distinguish probe sets representing genes with significant differential expression in the hearts from the different groups of mice, revealing genes that were up- and downregulated between the different groups studied. Among the identified differentially expressed genes, genes encoding actin cytoskeleton regulatory components were significantly differentially expressed in the hearts of WT mice expressing AAV-cofilin-1, similar to

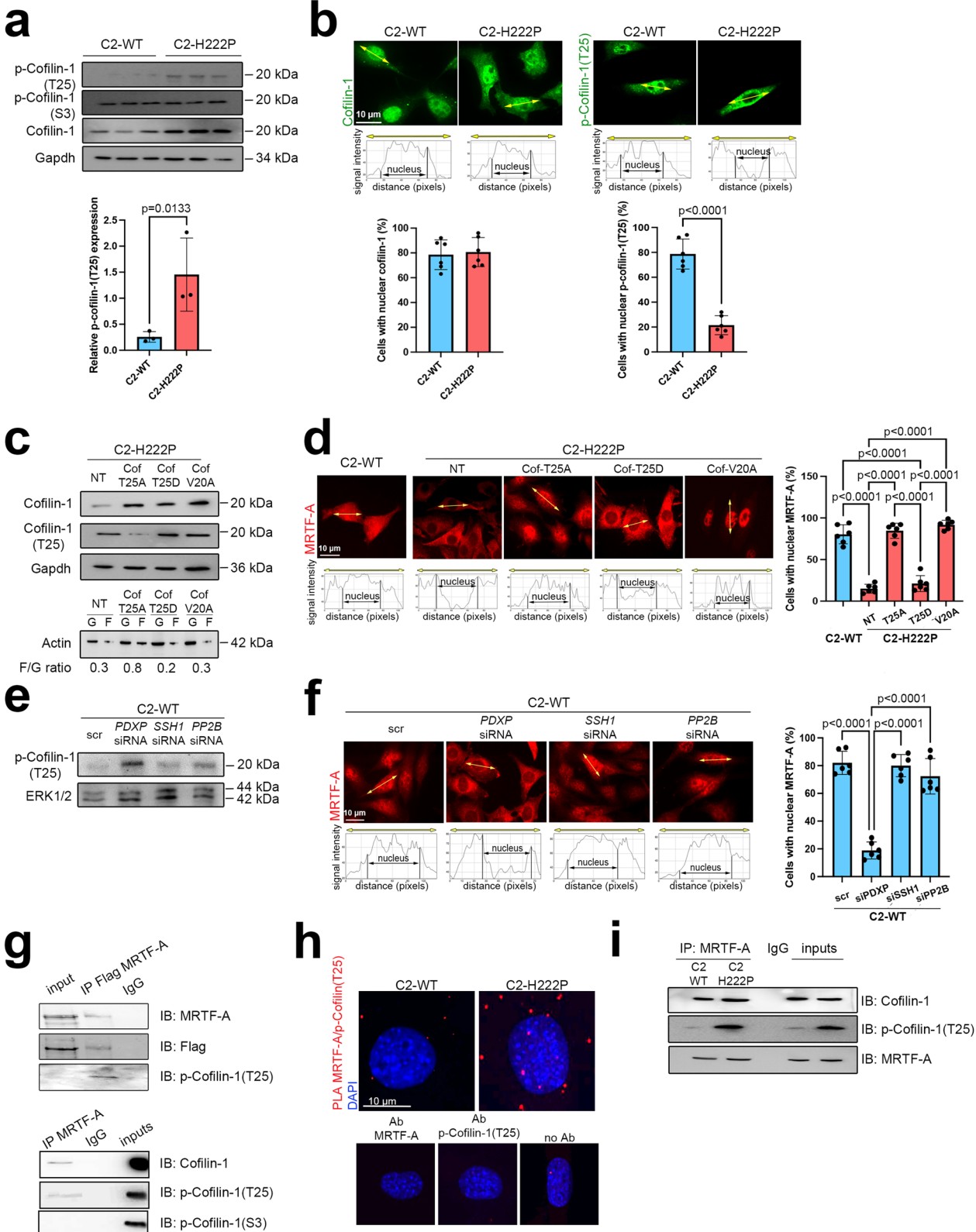

*Lmna*[p.H222P/H222P] mice (Tables S2, S3). These Gene Ontology classes were not altered in mice transduced with an AAV vector expressing non-phosphorylatable cofilin-1(T25A) or untransduced mice (Tables S2, S3). We next performed upstream regulator analysis to identify the cascade of upstream transcriptional regulators explaining the observed gene expression changes in WT mice transduced with or without an AAV vector expressing either cofilin-1 or cofilin-1(T25A). We found that SRF was significantly predicted to regulate the Gene Ontology classes that were enriched in WT mice expressing AAV-cofilin-1 and in *Lmna*[p.H222P/H222P] mice but not in WT mice expressing virus-encoded nonphosphorylatable cofilin-1(T25A) (Table S4). Moreover, many genes known to be regulated by SRF[46,48] were significantly downregulated in WT mice expressing AAV-cofilin-1 and in *Lmna*[p.H222P/H222P] mice as compared with WT mice expressing virus-encoded cofilin-1(T25A)

**Fig. 2 | Cofilin-1 phosphorylated on threonine 25 binds to MRTF-A and prevents its nuclear localization. a** Immunoblots showing total, phospho(S3) and phospho(T25)-cofilin-1 expression in C2-WT and C2-H222P cells. GAPDH was used as a loading control. Bar graph shows the quantification of phospho(T25)-cofilin-1 ($n = 3$ independent experiments, mean ± SD). **b** Representative immunofluorescence staining of cofilin-1 and phospho(T25)-cofilin-1 in C2-WT and C2-H222P cells. Scan line graphs represent the intensity of staining along the yellow arrows. Bar graph shows quantification of nuclear cofilin-1 ($n = 487$ cells, mean ± SD) and phospho(T25)-cofilin-1 staining ($n = 472$ cells, mean ± SD). **c** Immunoblots showing cofilin-1 and monomeric G-actin (G) vs. filamentous F-actin (F) expression in C2-H222P cells transfected with plasmids expressing nonphosphorylatable (T25A), phosphomimetic (T25D), and NES-mutated (V20A) forms of cofilin-1. GAPDH was used as a loading control. F/G ratios were calculated from $n = 3$. **d** Representative immunofluorescence staining and scan lines of MRTF-A in C2-WT and C2-H222P cells transfected with same plasmids as in (**c**). Bar graph shows the quantification of nuclear MRTF-A ($n > 200$ cells over 6 independent experiments, mean ± SD). **e** Immunoblots showing phospho(T25)-cofilin-1 expression in C2-WT cells transfected with siRNAs silencing PDXP, SSH1 or PP2B phosphatases. ERK1/2 was used as a loading control. **f** Representative immunofluorescence staining and scan lines of MRTF-A in C2-WT cells transfected with the same siRNAs as in (**e**). Bar graph shows the quantification of nuclear MRTF-A ($n > 200$ cells over six independent experiments, mean ± SD). **g** Immunoblots showing the interaction of phospho(T25)-cofilin-1 and MRTF-A. Top: proteins from C2C12 were immunoprecipitated with Flag antibody and immunoblotted with MRTF-A or phospho(T25)-cofilin-1 antibodies. Bottom: proteins from C2C12 were immunoprecipitated with MRTF-A antibody and immunnoblotted with cofilin-1, phospho(T25)-cofilin-1, or phospho(S3)-cofilin-1 antibodies. IgG was used as a negative control. **h** Representative micrographs from proximity ligation assay (PLA) between MRTF-A and phospho(T25)-cofilin-1 in C2-WT and C2-H222P cells. Nuclei were counterstained with DAPI. No positive PLA reactions (red dots) were observed for MRTF-A or phospho(T25)-cofilin-1 alone or for negative control without primary antibodies (no Ab). **i** Immunoblots showing the interaction of phospho(T25)-cofilin-1 and MRTF-A. Proteins from C2-WT ($n = 3$) and C2-H222P ($n = 3$) cells were immunoprecipitated with MRTF-A and immunoblotted with cofilin-1 or phospho(T25)-cofilin-1 antibodies. IgG was used as a negative control. For **g**–**i** representative of three independent repeats is shown. Statistics: for **a** and **b**, unpaired two-tailed *t*-test; for **d** and **f**, one-way ANOVA followed by Tukey's multiple comparison test. Source data are provided as a Source Data file.

(Table S5). Together, these results show that the expression of cofilin-1 phosphorylated on T25 affects SRF-mediated gene expression and leads to cardiomyopathy.

## α-Tubulin acetylation is decreased through SRF-mediated expression of *ATAT1* in muscle cells, mice, and human cardiac tissue expressing cardiomyopathy-causing mutant A-type lamins

SRF mediates the transcription of *ATAT1* encoding α-tubulin acetyltransferase 1 (α-TAT1)[49], which catalyzes the lysine (K)40 acetylation of α-tubulin[50,51], which in turn is specifically deacetylated by histone deacetylase-6 (HDAC6)[52,53] (schematically shown in Fig. 4a). We assessed whether SRF regulates α-tubulin acetylation by controlling *ATAT1* expression in cardiomyopathy caused by mutations in *LMNA*. To assess the impact of SRF downregulation on *ATAT1* transcription, we used a luciferase reporter plasmid with a canonical MRTF-A/SRF-CArG sequence upstream of a minimal *ATAT1* promoter (schematically shown in Fig. 4b). Indeed, we found that the transcriptional activity of the *ATAT1* promoter was reduced in C2-H222P cells compared with C2-WT cells (Fig. 4c). To demonstrate the causal relationship of MRTF-A/SRF on *ATAT1* transcription, we performed gain and loss-of-function experiments. For gain of function, we transfected C2C12 cells with a plasmid expressing constitutively active SRF-VP16 (a fusion protein of SRF and the viral VP16 transactivation domain) or with WT MRTF-A. For loss of function, cells were transfected with MRTFΔ100, an MRTF-A deleted in a region required for its nuclear import and transcriptional activity. SRF-VP16 and MRTF transfections increased *Atat1* along with *Srf* and *Acta2* expression. Concordantly, transfection with MRTFΔ100 had no effect on these three genes expression. These results confirm that MRTF-A/SRF axis regulates *Atat1* gene expression (Fig. 4d).

Moreover, the expression of α-TAT1 was downregulated in the hearts of *Lmna*[p.H222P/H222P] mice compared with those of control mice at 6 months of age (Fig. 4e). Considering the important role of microtubule mechanics in cardiomyocytes[33], we examined whether the K40 acetylation of α-tubulin, which promotes microtubule stability and resistance to mechanical breakage, was altered in cardiomyopathy caused by mutations in *LMNA* as a result of the decreased α-TAT1 expression. A comparison of cardiac protein extracts from WT and *Lmna*[p.H222P/H222P] mice by immunoblot revealed the downregulation of α-tubulin K40 acetylation in the hearts of mice with *Lmna* mutations (Fig. 4f). We confirmed this result via immunoblot and immunofluorescence in cardiomyocytes isolated from WT and *Lmna*[p.H222P/H222P] mice (Fig. 4g). To determine the role of α-tubulin K40 acetylation in human disease, we next analyzed cardiac tissue from explanted hearts from patients with cardiomyopathy caused by mutations in *LMNA*

following a heart transplant. We observed that the K40 acetylation of α-tubulin was decreased in the *LMNA*-mutated heart tissue compared with control heart tissue (Fig. 4h). Similar α-tubulin K40 acetylation profile was observed in immunoblot and immunofluorescence assays of cardiomyocytes derived from patient-specific human iPSCs (carrying *LMNA* mutations (Fig. 4i). These observations indicated that the acetylation of α-tubulin was abnormally downregulated in cardiomyopathy caused by mutations in *LMNA*. Further, our data support the hypothesis that decreased α-TAT1 expression is responsible for the reduced levels of tubulin acetylation in models expressing mutated Lamin A.

To confirm that the abnormal α-tubulin K40 acetylation profile was the consequence of the *LMNA* mutations and not a secondary effect of cardiac disease, we next used C2-WT and C2-H222P cells. We confirmed that the acetylation level of α-tubulin was lower in cells expressing mutated p.H222P Lamin A than in cells expressing WT Lamin A (Fig. 4j). Histone deacetylase 6 (HDAC6) is the major deacetylase responsible for removing the acetyl group of α-tubulin[52,53]. We did not report any modulation in HDAC6 expression neither in the *Lmna*-mutated heart tissue compared with control heart tissue from mice nor in C2-H222P cells compared with C2-WT cells (Fig. 4k). Together, these data indicate that A-type lamin variants negatively regulate the acetylation of α-tubulin in heart tissues and in cultured cells, mainly through α-TAT1 expression.

## α-Tubulin acetylation mediates cardiac function

To further evaluate the importance of the effects of α-tubulin K40 acetylation on cardiac function in vivo, we studied mice bearing a targeted deletion of *Atat1*[54]. The cardiac tissues of homozygous *Atat1* knockout mice (*Atat1*[-/-]) did not exhibit detectable levels of acetylated α-tubulin as compared with WT mice (Fig. 5a). We next analyzed *Atat1*[-/-] mice by echocardiography to test their cardiac function. Compared to WT mice, *Atat1*[-/-] mice had significantly increased left ventricular end-diastolic and end-systolic diameters, demonstrating left ventricle dilation, starting as early as 3 months of age (Table 1, Fig. 5b). We further found significantly decreased fractional shortening in 3-month-old *Atat1*[-/-] mice (Table 1, Fig. 5b). Our results revealed a previously unnoticed cardiac dilation phenotype of *Atat1*[-/-] mice.

Given that the deacetylation of α-tubulin is driven by HDAC6[52,53], and to confirm the role of α-tubulin K40 acetylation on cardiac function, we studied by echocardiography the cardiac function of WT mice transduced with or without an AAV vector expressing HDAC6 (AAV-HDAC6). Compared with WT mice, mice transduced with AAV-HDAC6 had decreased acetylation levels of α-tubulin (Fig. 5c). This is ensuing significantly increased left ventricular end-systolic diameters

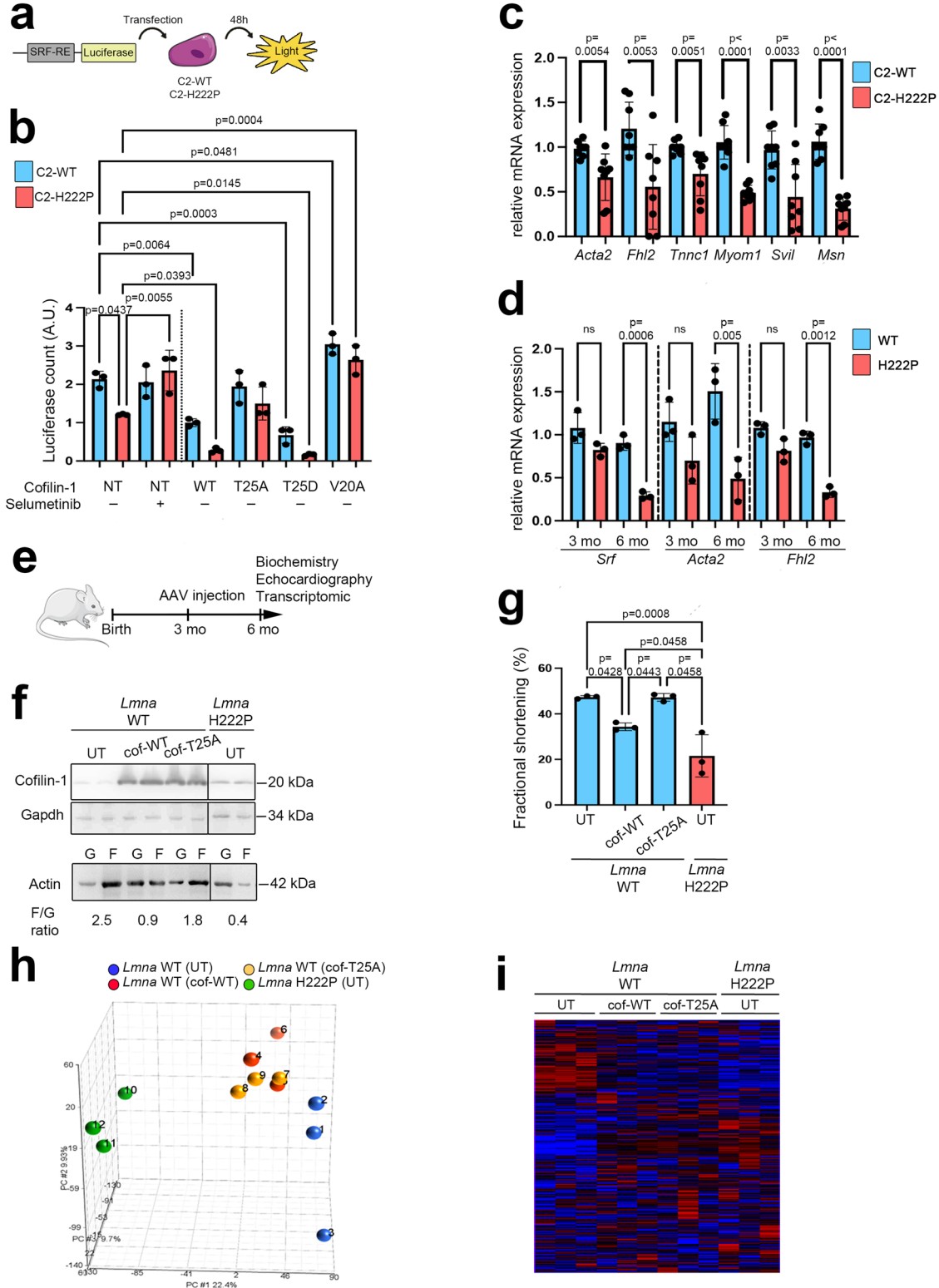

and decreased fractional shortening (Table 2, Fig. 5d). Taken together, these data highlight the importance of α-tubulin K40 acetylation for cardiac function.

We next assessed whether increasing the acetylation levels in *Lmna*[p.H222P/H222P] mice with the HDAC6 inhibitor tubastatin A[55] could improve the cardiac function in these animals. We treated 2-month-old *Lmna*[p.H222P/H222P] mice with tubastatin A for 1 month and then performed echocardiography. Tubastatin A treatment increased the level of α-tubulin acetylation in hearts of *Lmna*[p.H222P/H222P] mice as compared with

DMSO treatment (Fig. 5e) without altering other HDAC expression or acetylation of histone H3 (Fig. S1). Moreover, the left ventricular end-systolic diameter was significantly lowered and fractional shortening was significantly improved in *Lmna*[p.H222P/H222P] mice treated with tubastatin A compared with DMSO treated (Table 3, Fig. 5f). In vitro, treatment with tubastatin A of cardiomyocytes derived from iPS (iPS-CM) from a patient with p.H222P *LMNA* mutation was able to increase α-tubulin acetylation (Fig. 5g) and decrease contraction frequency (Fig. 5h) compared with untreated cells. Tubastatin A treatment was

**Fig. 3 | Phospho(T25)-cofilin-1 impedes SRF regulation and alters cardiac function in mice. a** Schematic representation of the luciferase assay for assessing SRF activity. **b** Bar graph showing quantification of luciferase activity in C2-WT and C2-H222P cells overexpressing SRF response element and transfected or not (NT) with plasmids expressing different mutated cofilin-1 proteins. Results after treatment with selumetinib are also shown (*n* = 3 independent experiments, mean ± SD). **c** mRNA expression of SRF target-genes: *Acta2, Fhl2, Tnnc, Myom1, Svil*, and *Msn* in C2-H222P and C2-WT cells (*n* = 6 independent experiments, mean ± SD). **d** mRNA expression of *Srf, Acta2* and *Fhl2* in hearts of 3 and 6-month-old male *Lmna*p.H222P/H222P (H222P) and WT mice (*n* = 3 mice per condition, mean ± SD). **e** Schematic representation of methodological protocol. WT mice were injected with AAV vectors encoding either WT or nonphosphorylable cofilin-1(T25A) at 3 months of age. Biochemistry, echocardiography, and transcriptomic analysis were performed at 6 months of age. Subsequent data showed in **f**–**i**, are obtained from the hearts of

6-month-old male *Lmna* WT mice transduced with AAV vectors encoding cofilin-1 or cofilin-1(T25A) and *Lmna* H222P mice (*n* = 3 mice per condition, mean ± SD). **f** Immunoblots showing cofilin-1 (top) and monomeric G-actin (G) and filamentous F-actin (F) (bottom) expression. GAPDH was used as the loading control. *Lmna* H222P mice are shown as controls, and UT indicates untransduced. **g** Bar graph showing the left ventricular fraction shortening (FS) values. **h** Principal component analysis (PCA) of the Affymetrix probe sets. **i** Unsupervised hierarchical clustering of the Affymetrix probe set results. Statistics: for **b** and **g**, one-way ANOVA followed by Tukey's multiple comparison test; for **c** and **d**, unpaired two-tailed *t*-test. Source data are provided as a Source Data file. For **a** and **e**, parts of the figure were drawn by using pictures from Servier Medical Art. Servier Medical Art by Servier is licensed under a Creative Commons Attribution 3.0 Unported License (https://creativecommons.org/licenses/by/3.0/).

also able to ameliorate the contraction profile of iPS-CM carrying *LMNA* p.H222P mutation (Fig. S2), suggesting an increased contraction force. This reflects the previously reported nonreciprocal relationship between force and contraction frequency in cardiac muscle cells[56–58]. These data demonstrate that α-tubulin acetylation plays key role in the development of cardiac dysfunction in mice with cardiomyopathy caused by mutations in *LMNA*.

One of the common cellular phenotypes observed in cardiomyopathy caused by mutations in *LMNA* is an abnormal elongation of nuclei in cardiomyocytes[3,4,59,60]. Mean length of cardiomyocyte nuclei in hearts of *Lmna*p.H222P/H222P mice treated with tubastatin A was significantly longer, as observed in untreated *Lmna*p.H222P/H222P mice, than in WT mice (Fig. S3a and b). Nuclei of cardiomyocytes in hearts of *Atat1*[-/-] mice, as well as WT mice transduced with AAV-HDAC6, had an overall shape that was "rounded", similar to WT mice (Fig. S3a and b). This suggests that α-tubulin K40 acetylation does not have a role in the shape of the nuclei.

## α-Tubulin acetylation mediates Cx43 localization in cellular and mouse models expressing cardiomyopathy-causing mutant A-type lamins

The development of cardiomyopathy caused by mutations in *LMNA* was previously correlated with the remodeling of the gap junction protein Cx43 and characterized by a loss of localization at intercalated discs and redistribution to the lateral sides of cardiomyocyte membranes (i.e., "lateralization") in the hearts of *Lmna*p.H222P/H222P mice[9,61]. The trafficking of Cx43 and its correct localization at intercalated discs in the heart is regulated in part by the microtubule network[9,62,63]. We, therefore, suspect that the changes in α-tubulin acetylation might explain the Cx43 phenotype in cardiomyopathy caused by mutations in *LMNA*.

First, immunofluorescence analyses confirmed that Cx43 underwent extensive remodeling in isolated cardiomyocytes (Fig. 6a) and whole hearts (Fig. 6b) from *Lmna*p.H222P/H222P mice at an earlier stage (3 months) of disease. To further study the mechanisms underlying this Cx43 remodeling, we used C2-H222P and C2-WT cells. We first assessed the localization of Cx43 in C2-H222P and C2-WT cells. While Cx43 was localized at the cell-to-cell junction in C2-WT cells, it was mostly localized in the cytoplasm of C2-H222P cells (Fig. 6c). Similar alterations were observed when C2-H222P cells were cultured in 3D conditions (i.e., as spheroids; Fig. 6d).

Considering that the levels of K40-acetylated α-tubulin were decreased in the presence of mutated A-type lamin, we next asked whether altering the α-tubulin acetylation levels could affect Cx43 localization in cells. To test the effect of decreased tubulin acetylation, we knocked down Atat1 in C2C12 cells using siRNA (Fig. 6e, top panel), which resulted in the mislocalization of Cx43 (Fig. 6e, bottom panel). To further confirm that the acetylation of α-tubulin was critical for Cx43 localization, we then treated C2-H222P cells with tubastatin A, which increased the levels of acetylated α-tubulin in a dose-dependent

manner (Fig. 6f, top panel) and restored the normal localization of Cx43 (Fig. 6f, bottom panel). Together, these results demonstrate that the acetylation of α-tubulin is essential for the correct localization of Cx43 in muscle cells.

The trafficking of Cx43 along microtubules is driven in part by motor proteins from the kinesin-1 family[64]. It has also been shown that α-tubulin acetylation positively influences motor protein binding to microtubules and has a role in intracellular trafficking[65]. To determine if active, microtubule-based transport is important for Cx43 localization, we treated cells with adenosine 5′-(β,γ-imido)triphosphate (AMP-PNP), a non-hydrolysable analog of ATP that inhibits motility of kinesin motor proteins that drive intracellular transport. Following AMP-PNP treatment, Cx43 was misplaced within the cytoplasm of C2-WT myoblasts (Fig. S4a) and cardiomyocytes isolated from WT mice (Fig. S4b). To test the role of kinesin in this process more directly, we transfected C2-WT myoblasts with siRNA against Kif5B, the kinesin motor suggested to be responsible for Cx43 trafficking (Fig. S4c). In cells depleted of Kif5B, Cx43 was localized in the cytosol rather than at cell-cell junctions (Fig. S4d). This demonstrates an important role of active microtubule-based transport in the trafficking and localization of Cx43 to cell-cell junctions. This process is perturbed in the presence of mutated A-type lamins. To confirm the relevance of this mechanism in vivo, we performed immunostaining of Cx43 of cardiac sections. We observed Cx43 remodeling and redistribution to the lateral sides in *Atat1*[-/-] mouse cardiomyocytes, similar to the phenotype observed in *Lmna*p.H222P/H222P hearts (Fig. 6g top panel, 6h). Similarly, Cx43 was relocated to the lateral sides in cardiomyocytes from WT mice transduced with an AAV-HDAC6 (Fig. 6g middle panel, 6h). Strikingly, Cx43 localization at intercalated discs in the hearts of *Lmna*p.H222P/H222P mice was restored by treatment with tubastatin A (Fig. 6g bottom panel, 6h). This demonstrates that α-tubulin acetylation plays a key role in the trafficking of Cx43 to intercalated discs.

Given that gap junctions are required to ensure a coordinated contraction and pumping activity in heart[66–68], we anticipated to prevent conduction defects (prolonged QRS complex duration) in tubastatin A-treated *Lmna*p.H222P/H222P mice. One-month tubastatin A treatment failed to shorten QRS duration in 3-month-old mice (Table S6, Fig. S5). Conversely, we expected to find conduction defects in *Atat1*[-/-] mice as well as in AAV-HDAC6 transduced WT mice. Surprisingly, we did not report any conduction defects (Table S6, Fig. S5). This could be explained in part by the fact that C57BL/6J genetic background mice are resistant to arrhythmia[69–71]. Also, abnormal tissue architecture due to increased fibrosis may have synergistic effects together with lateralization of Cx43 and lead to conduction slowing. We then performed Sirius red staining of collagen fibrils on heart sections of *Atat1*[-/-] and *Lmna*p.H222P/H222P mice treated or not with tubastatin A. As previously reported, *Lmna*p.H222P/H222P mice displayed significantly increased cardiac fibrosis[38] as compared with WT mice (Fig. 6i). Treatment with tubastatin A failed to prevent collagen fibrils accumulation (Fig. 6i). On the opposite, heart sections from *Atat1*[-/-]

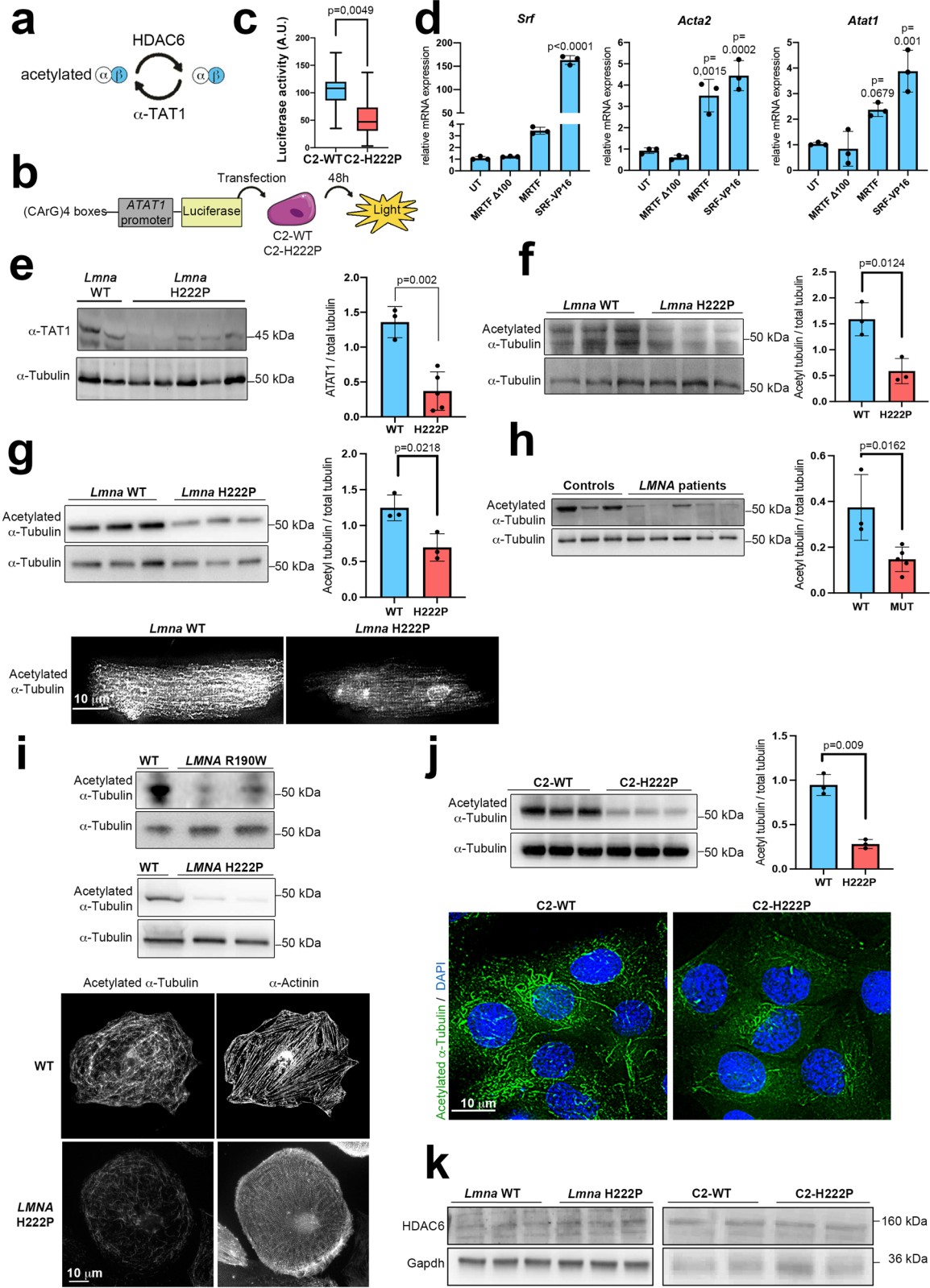

were similar to WT (Fig. 6i), meaning that lack of acetylated α-tubulin does not trigger cardiac fibrosis.

## Discussion

Dilated cardiomyopathy caused by mutations in *LMNA* is associated with both actin[7,8] and microtubule[9,10] network alterations. We aimed to decipher the mechanisms underlying the possible interplay between these two cytoskeletal components in this pathological context. Here, we uncovered a mechanism through which actin-microtubule crosstalk plays a major physiological role in cardiac function (illustrated in Fig. 7). Specifically, we found that cofilin-1 phosphorylation on T25 regulates the MRTF-A/SRF axis in cells expressing *LMNA* mutations. In this mechanism, phospho(T25)-cofilin-1 binds to MRTF-A, preventing its translocation to the nucleus, and thus blunts the transcription

**Fig. 4 | α-Tubulin acetylation is decreased through SRF-mediated expression of *ATAT1* in cells, and cardiac tissue expressing cardiomyopathy-causing mutant A-type lamins. a** Schematic view of the enzymes responsible for acetylation and deacetylation of α-tubulin. **b** Experimental protocol for measuring SRF transcriptional activity based on *ATAT1* expression by luciferase assay. **c** Box plot showing the quantification of luciferase activity after C2-WT and C2-H222P cells were transfected with the construction described in (**b**) ($n = 3$ independent experiments; whiskers min to max, line in the middle of the box is plotted at the median). **d** Bar graph showing mRNA relative expression of *Srf*, *Acta2*, and *Atat1* in C2C12 transfected with plasmids expressing inactive MRTF-A (MRTFΔ100), WT MRTF-A (MRTF), or constitutively active SRF (SRF-VP16) ($n = 3$ independent experiments, mean ± SD). **e** Left: immunoblot showing α-TAT1 and α-tubulin expression in hearts of 6-month-old male WT and *Lmna*[p.H222P/H222P] (H222P) mice. Right: bar graph showing the quantification of α-TAT1 ($n = 3$ WT and $n = 5$ H222P, mean ± SD). **f–j** Top left: immunoblot showing expression of acetylated and total α-tubulin, top right: bar graph showing the quantification of acetylated α-tubulin normalized to total α-tubulin (**f**) in the hearts of 3-month-old male WT and *Lmna*[p.H222P/H222P] (H222P) mice ($n = 3$ mice per condition, mean ± SD). **g** in adult cardiomyocytes isolated from 3-month-old male WT and *Lmna*[p.H222P/H222P] (H222P) mice ($n = 3$ mice per condition,

mean ± SD). Bottom: representative immunofluorescence staining of acetylated α-tubulin (**h**) in explanted heart tissue from control individuals ($n = 3$) and patients carrying *LMNA* point mutations ($n = 5$) (mean ± SD). **i** in cardiomyocytes derived from iPS cells from a control (WT), a patient carrying *LMNA* p.R190W mutation (upper panel) and a patient carrying *LMNA* p.H222P mutation (lower panel). Bottom: representative immunofluorescence staining of acetylated α-tubulin in cardiomyocytes derived from iPS WT and H222P. α-Actinin is shown as cardiomyocyte differentiation marker. **j** In C2-WT and C2-H222P cells ($n = 3$ independent experiments, mean ± SD). Bottom: representative immunofluorescence staining of acetylated α-tubulin. **k** Immunoblot showing expression of HDAC6 in hearts of 6-month-old male WT mice and *Lmna*[p.H222P/H222P] (H222P) mice ($n = 3$ mice per condition) as well as in C2-WT and C2-H222P cells (a representative of three independent repeats is shown). Statistics: for **c**, **e–h**, and **j**, unpaired two-tailed *t*-test; for **d**, one-way ANOVA followed by Tukey's multiple comparison test. Source data are provided as a Source Data file. For **b**, parts of the figure were drawn by using pictures from Servier Medical Art. Servier Medical Art by Servier is licensed under a Creative Commons Attribution 3.0 Unported License (https://creativecommons.org/licenses/by/3.0/).

factor activity of SRF. This in turn leads to the loss of α-tubulin K40 acetylation due to the downregulation of α-TAT1 expression and thereby disrupts the localization of Cx43 in cardiomyocytes, participating in the development of cardiomyopathy. Further, we showed that elevation of α-tubulin acetylation levels with tubastatin A restored the correct localization of Cx43 at intercalated discs and improved cardiac function in a mouse model of cardiomyopathy caused by *LMNA* mutations.

Cofilin-1 is an essential protein that maintains the myofilament architecture needed for the mechanical properties of sarcomeres[7,43], cell motility[72], and nuclear architecture[73]. Cofilin-1 phosphorylation controls the interactions of cofilin-1 and actin and thus the mechanics of the actin regulatory machinery[74]. The phosphorylation of cofilin-1 on Ser3 by TESK and LIMK was previously described to markedly attenuates cofilin-1/actin interactions[16,75,76], while dephosphorylation at this site restores the affinity of cofilin-1 for actin[77]. Cofilin phosphocycling at this residue clearly drives a diverse variety of actin-dependent processes, ranging from actin-based motility[78] to the pathogenesis of neurodegenerative diseases[79,80]. Here, we uncover a novel role of cofilin-1 through phosphorylation on Thr25 by ERK1/2 (kinase) and dephosphorylation by PDXP (phosphatase). The control of cofilin-1 phosphorylation on Thr25, in addition to its effect on actin dynamics, directly regulates MRTF-A/SRF signaling activity in cardiac cells and an imbalance in this regulation ultimately leads to cardiac disease associated with mutations in *LMNA*. Our results also broaden our knowledge of diseases related to cofilin-1, as we demonstrate that cofilin-1 participates in the pathogenesis of left ventricular dysfunction through the transcriptional control of *ATAT1* driving α-tubulin acetylation.

Previous work showed that activation of ERK1/2 signaling induces actin depolymerization via cofilin-1 phosphorylation at T25 and promotes the disassembly of F-actin filaments into G-actin monomers[7]. We now show that phospho(T25)-cofilin-1 directly binds to MRTF-A and sequesters it in the cytoplasm, thereby inhibiting SRF transcriptional activity. MRTF-A physically associates with SRF to synergistically activate the transcription of a subset of CArG box-containing genes[17,19,81,82], including those encoding actin microfilament effectors such as cofilin-1[83,84]. SRF plays a crucial role in cardiac maturation[85] and function[86], and the cardiac-specific deletion of SRF in adult mice results in cardiomyopathy[23]. We show that the expression of the acetylating enzyme α-TAT1, which drives the acetylation of α-tubulin, is controlled by the activity of SRF and plays a role in cardiac function. Our findings provide a novel link between impaired SRF signaling and the perturbation of cellular functions that lead to cardiac defects in cardiomyopathy caused by mutations in

*LMNA*. Given that SRF transcriptional activity is dependent on actin cytoskeleton dynamics[24], it is becoming increasingly clear that the two cytoskeletal systems work together in core cellular processes and that their functional dynamics are often intimately interwoven[11]. Recent research shows that actin and microtubule networks engage in a variety of types of crosstalk that are important for biological processes such as cell migration, axonal specification, cell polarity, or cell division[11]. Our work shows that changes in the regulation of a single factor, cofilin-1, trigger a cascade of molecular events impacting both actin and microtubule dynamics, further highlighting that the cardiac cytoskeleton is not a set of individual elements but rather a cooperative system in which key elements function synergistically in a precise and highly complementary manner.

Altered Cx43 localization and disorganized gap junction coupling contribute to the pathogenesis of cardiomyopathy caused by mutations in *LMNA*[9,61]. Tubulin acetylation is known to influence the binding of motor proteins to microtubules and to play a role in intracellular Cx43 trafficking[65,87]. Changes in the localization and regulation of Cx43 have been described in many forms of cardiac diseases including heart failure[88–90]. However, the molecular mechanisms of gap junction remodeling are not yet known, and their elucidation is of paramount importance for the development of therapies aimed at improving gap junction coupling during disease. Microtubules have a well-studied role in the trafficking of Cx43 to the plasma membrane[62,91,92], and the implication of the microtubule network in heart disease is solidly established. Recent findings highlighted the role of tubulin detyrosination in cardiomyopathy, as a perturbation of this post-translational modification was shown to correlate with functional decline in the heart[32,33]. Here, we demonstrated that the remodeling of cardiac Cx43 in cardiomyopathy caused by mutations in *LMNA* occurred due to the decreased acetylation of α-tubulin, thus establishing this tubulin post-translational modification as a cause of the cardiac disease phenotype. Post-translational modifications can play important roles in controlling the stability and function of microtubules. Most of tubulin post-translational modifications (e.g., detyrosination, polyglutamylation, and polyglycylation) alter residues within the C-terminal tail of tubulin that extends from the surface of the microtubule. However, α-tubulin K40 acetylation is the unique tubulin post-translational modification that localizes to the inside of the microtubule. Acetylation of α-tubulin K40 was found to be stable, long-lived microtubule subpopulations[30,93]. However, it still remains controversial whether the relationship between α-tubulin K40 acetylation and microtubule stabilization is correlative or causative. How decreased tubulin acetylation alters the overall architecture of the microtubule cytoskeleton

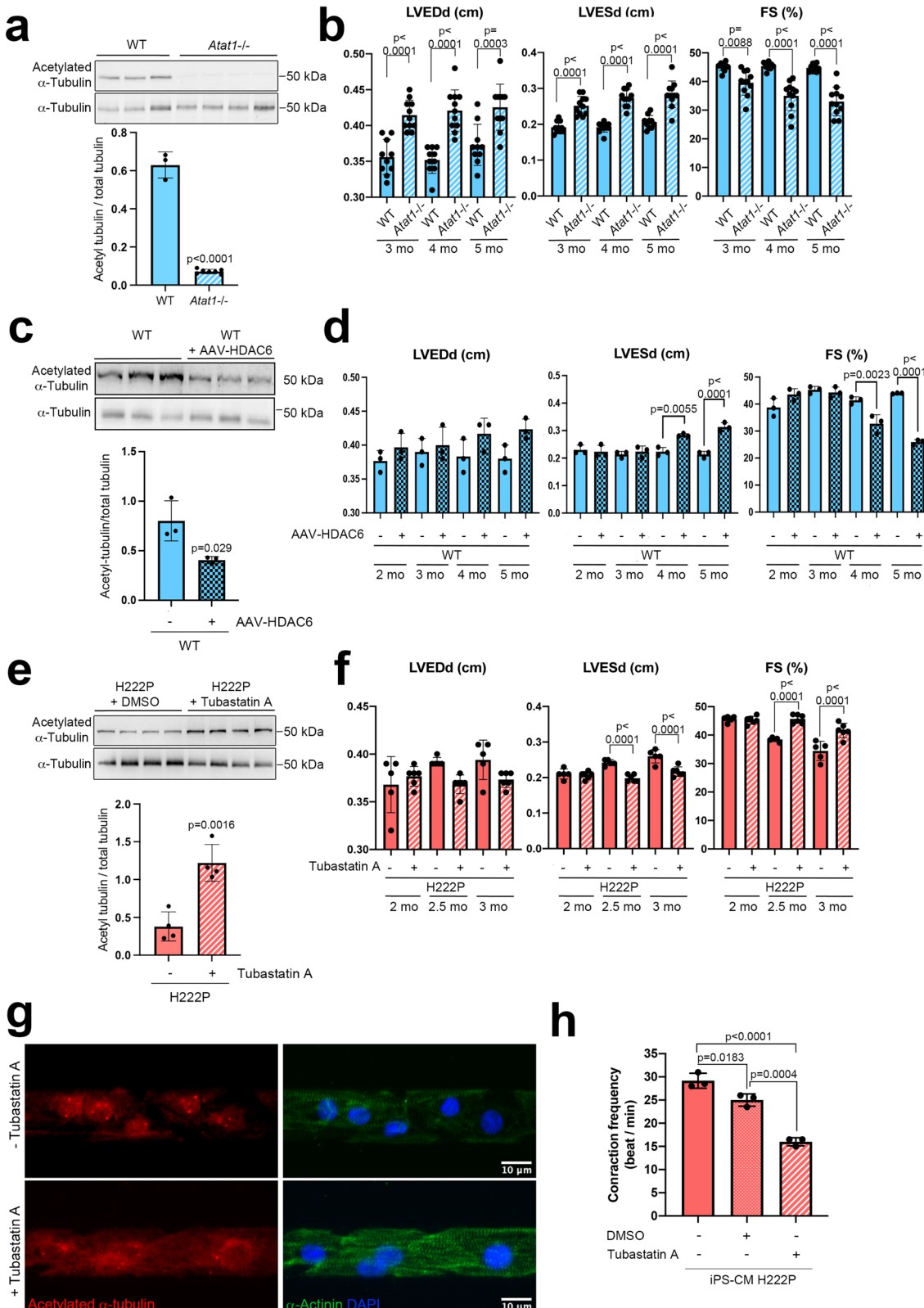

and disturbs Cx43 targeting intercalated discs remains to be explored. However, acetylated α-tubulin stabilizes microtubules[94] and a decreased acetylation could alter microtubules longitudinal alignment within cardiomyocytes that in turn, might affect Cx43 localization (illustrated in Fig. 7). Moreover, our observations could have broader implications for other physiologically important intracellular trafficking events in cardiomyocytes. For example,

voltage-gated potassium channels, which are involved in the excitability of cardiac muscle cells and modulate contraction and relaxation, are transported along microtubules[95–97]. It will thus be exciting to study the trafficking of other channels in cardiomyopathy caused by mutations in *LMNA*, which may help to identify novel therapeutic targets for cardiac disorders that arise from the deficient delivery of ion channels to the plasma membrane.

**Fig. 5 | α-Tubulin acetylation mediates cardiac function in *Atat1* knock-out mice and in mice expressing cardiomyopathy-causing mutant A-type lamins.** **a**, **c**, **e** Top: Immunoblots showing the cardiac protein expression of acetylated and total α-tubulin **a** in 6-month-old male *Atat1*[+/+] (WT) mice (*n* = 3) and *Atat1* knockout mice (*Atat1*[−/−]) (*n* = 5) **c** in 5-month-old male WT mice (*n* = 3) and WT mice transduced with AAV expressing HDAC6 (*n* = 3), and **c** in 3-month-old male *Lmna*[p.H222P/H222P] (H222P) mice treated with tubastatin A for one month (*n* = 4) compared with mice treated with DMSO (*n* = 4). **a**, **c**, **e** Bottom: bar graph showing the quantification of acetylated α-tubulin normalized to total α-tubulin (mean ± SD). **b**, **d**, **f** Bar graphs showing the left ventricular end-diastolic diameters (LVEDd), end-systolic diameters (LVESd) and fraction shortening (FS) (mean ± SEM) **b** in *Atat1*[+/+] (WT) mice (*n* = 10) and *Atat1* knockout mice (*Atat1*[−/−]) (*n* = 11) from 3 to 6 months of age, **d** in 5-month-old male WT mice (*n* = 3) and WT mice transduced

with AAV expressing HDAC6 (*n* = 3) and **f** in 3-month-old male *Lmna*[p.H222P/H222P] (H222P) mice treated for one month with tubastatin A (*n* = 6) compared to mice treated with DMSO (*n* = 5). **g** Immunofluorescence staining of acetylated α-tubulin and α-actinin in cardiomyocytes derived from iPS cells from a patient carrying the *LMNA* p.H222P mutation, 40 days post-differentiation, treated (lower panel) or not (upper panel) with 3 µM of tubastatin A for 24 h. α-Actinin is shown as cardiomyocyte differentiation marker. **h** Contraction frequency of cardiomyocytes derived from iPS cells from a patient carrying the *LMNA* p.H222P mutation, 40 days post-differentiation, treated or not with 3 µM of tubastatin A for 24 h (*n* = 3 technical replicates) (mean ± SD). Statistics: for **a**, **c**, **e** unpaired two-tailed *t*-test; for **b**, **d**, **f**, **h** one-way ANOVA followed by Tukey's multiple comparison test. Source data are provided as a Source Data file.

**Table 1 | Echocardiographic data for *Atat1*[+/+] (WT) and *Atat1*[−/−] (Atat1 KO) mice at 3; 4 and 5 months of age**

|  | WT | Atat1 KO | WT | Atat1 KO | WT | Atat1 KO |
|---|---|---|---|---|---|---|
| Age | 3 months | 3 months | 4 months | 4 months | 5 months | 5 months |
| *n* | 10 | 11 | 10 | 11 | 10 | 11 |
| Time (ms) | 93.3 ± 1.2 | 94.8 ± 1.2 | 92.3 ± 1.4 | 91.5 ± 1.4 | 90.7 ± 1.2 | 95.3 ± 1.3 |
| Heart rate (bpm) | 644.1 ± 8.9 | 635.0 ± 8.8 | 650.9 ± 9.9 | 657.6 ± 10.11 | 662.5 ± 9.4 | 630.4 ± 8.5 |
| Cardiac output (l/min) | 0.064 ± 0.004 | 0.087 ± 0.003*** | 0.061 ± 0.03 | 0.084 ± 0.004 *** | 0.072 ± 0.003 | 0.081 ± 0.004 |
| IVSd (cm) | 0.061 ± 0.001 | 0.067 ± 0.001* | 0.062 ± 0.001 | 0.069 ± 0.001* | 0.067 ± 0.002 | 0.068 ± 0.001 |
| LVDd (cm) | 0.35 ± 0.01 | 0.41 ± 0.01**** | 0.35 ± 0.01 | 0.42 ± 0.01**** | 0.37 ± 0.01 | 0.42 ± 0.01*** |
| LVPWd (cm) | 0.058 ± 0.003 | 0.059 ± 0002 | 0.065 ± 0.001 | 0.066 ± 0.004 | 0070 ± 0,004 | 0.068 ± 0.003 |
| IVSs (cm) | 0.11 ± 0.004 | 0.11 ± 0.002 | 0.11 ± 0.003 | 0.11 ± 0.003 | 0.11 ± 0.003 | 0.10 ± 0.004 |
| LVDs (cm) | 0.19 ± 0.005 | 0.25 ± 0.008**** | 0.19 ± 0.004 | 0.27 ± 0.008**** | 0.21 ± 0.006 | 0.28 ± 0.011**** |
| LVPWs (cm) | 0.11 ± 0.002 | 0.11 ± 0.004 | 0.11 ± 0.001 | 0.10 ± 0.005 | 0.12 ± 0.004 | 0.10 ± 0.004** |
| LVED vol (ml) | 0.11 ± 0.008 | 0.17 ± 0.007**** | 0.11 ± 0.005 | 0.19 ± 0.008**** | 0.13 ± 0.008 | 0.19 ± 0.012*** |
| LVES vol (ml) | 0.021 ± 0.002 | 0.042 ± 0.003** | 0.019 ± 0.001 | 0.053 ± 0.004 **** | 0.023 ± 0,002 | 0.063 ± 0.007 **** |
| EF (%) | 82.84 ± 0.27 | 76.26 ± 1.55* | 82.71 ± 0.37 | 70.57 ± 2.16**** | 82.03 ± 0.37 | 67.97 ± 2.04**** |
| FS (%) | 45.26 ± 0.46 | 39.57 ± 1.31** | 45.41 ± 0.38 | 35.02 ± 1.57**** | 44.75 ± 0.37 | 33.04 ± 1.51**** |
| LV eject vol (ml) | 0.09 ± 0.008 | 0.13 ± 0.005*** | 0.09 ± 0.003 | 0.13 ± 0..007*** | 0.11 ± 0.007 | 0.13 ± 0.006 |
| h/r | 0.34 ± 0.012 | 0.31 ± 0.009 | 0.36 ± 0.006 | 0.31 ± 0.011** | 0.37 ± 0.007 | 0.32 ± 0.009* |

Statistics: one-way ANOVA followed by Tukey's multiple comparison test. Values are means ± SEM.
*IVS* interventricular septum, *LVD* left ventricular diameter, *LVPW* left ventricular posterior wall, *LVED* left ventricular end diastolic, *LVES* left ventricular end systolic, *EF* ejection fraction, *FS* fractional shortening, *s* systole, *d* diastole.
*$p \leq 0.05$, **$p \leq 0.01$, ***$p \leq 0.001$, and ****$p \leq 0.0001$ between Atat1[−/−] and wild type mice.

We showed that rescuing cardiac Cx43 localization with HDAC6 inhibitor is a straightforward therapeutic strategy. Aberrant tubulin acetylation profiles are observed in a broad spectrum of diseases[94,98–100], which has led to using HDAC6 inhibition as a therapeutic approach in preclinical studies[101–105]. General HDAC inhibitors lead to the deacetylation of histones, modify gene transcription, and increase the acetylation of many non-histone proteins[106,107]. HDAC6 does not act on histones and its major substrate is α-tubulin[108]. Acetylated α-tubulin occurs on polymerized microtubules and affects microtubule dynamics and stability[31]. Hence, the finding that α-tubulin acetylation in the heart is involved in cardiomyopathy, and that specific inhibition of HDAC6 has beneficial effects, opens up new therapeutic possibilities. Furthermore, while α-tubulin is the main target of HDAC6, this latter has other substrates including cortactin, ERK1, and Smad3 that have been shown to be dysregulated in *LMNA*-cardiomyopathy[39,109–112]. It has also been reported that HDAC6 inhibition has a cardioprotective effect through autophagy induction, a mechanism also affected in *LMNA*-cardiomyopathy[98,113]. A synergistic effect through α-tubulin, cortactin, ERK, and Smad signaling and autophagy induction might be highly beneficial in *LMNA*-cardiomyopathy.

Most of the promising HDAC inhibitors target multiple HDAC isoforms. These pan-HDAC inhibitors are often driving several undesirable side effects, which limits the clinical utility of these molecules. Selective HDAC6 inhibitor tubastatin A[55] may exhibit reduced side

effects compared to the pan-HDAC inhibitors. However, tubastatin A belongs to the hydroxamate class of molecules and displays mutagenicity/genotoxicity and poor oral delivery, thus hindering its development for clinical use[114]. Therefore, the development of a highly potent and selective HDAC6 inhibitor with good bioavailability and without genotoxicity is highly desired. Access to such HDAC6 inhibitor would enable us to study the effects of HDAC6 inhibition in cardiomyopathy caused by mutations in *LMNA* in preclinical in vivo models for translation to clinical trials. Among HDAC6 inhibitors, only ricolinostat (ACY-1215, ricolinostat) and citarinostat (ACY-241, citarinostat) are currently evaluated in clinical trials for several forms of cancer[35,109,115]. Given that HDAC6 inhibitors have lower adverse effects than pan-HDAC inhibitors, such compounds may be important in the array of therapeutic approaches for cardiomyopathy. This finding is notable because despite the growing evidence that microtubule-targeting strategies can be employed to slow or even halt the progression of inherited cardiomyopathies[9,32,34], clinical translation has thus far been hindered by the lack of available practical therapeutic targets. Our recent work showed that increasing α-tubulin K40 acetylation by augmenting microtubule stability using paclitaxel, could also be potentially translatable into therapy for cardiomyopathy caused by mutations in *LMNA*[9]. It has been shown that paclitaxel can be safely administered in patients with cardiac dysfunction[116]. Other microtubule-stabilizing agents have been described and should also

**Table 2 | Echocardiographic data for wild-type (WT) mice transduced or not with AAV expressing HDAC6 (AAV-HDAC6) at 2, 3, 4 and 5 months of age**

| | WT | AAV-HDAC6 | WT | AAV-HDAC6 | WT | AAV-HDAC6 | WT | AAV-HDAC6 |
|---|---|---|---|---|---|---|---|---|
| Age | 2 months | 2 months | 3 months | 3 months | 4 months | 4 months | 5 months | 5 months |
| n | 3 | 3 | 3 | 3 | 3 | 3 | 3 | 3 |
| Time (ms) | 89.3±0.68 | 93.4±2.3 | 93.4±2.1 | 92.9±1.2 | 92.5±0.4 | 91.9±0.9 | 94.4±0.6 | 93.5±5.6 |
| Heart rate (bpm) | 671.7±5.2 | 643.3±15.79 | 642.8±14.61 | 645.6±9.03 | 643.9±4.65 | 653.3±7.04 | 635.95±4.16 | 646.3±37.64 |
| Cardiac output (l/min) | 0.066±0.003 | 0.083±0.008 | 0.08±0.005 | 0.083±0.008 | 0.073±0.008 | 0.076±0.008 | 0.070±0.005 | 0.070±0.000 |
| IVSd (cm) | 0.063±0.003 | 0.066±0.003 | 0.070±0.000 | 0.066±0.003 | 0.070±0.005 | 0.066±0.003 | 0.073±0.003 | 0.063±0.003 |
| LVDd (cm) | 0.37±0.008 | 0.39±0.01 | 0.39±0.01 | 0.40±0.01 | 0.39±0.02 | 0.041±0.011 | 0.38±0.01 | 0.42±0.008 |
| LVPWd (cm) | 0.06±0.000 | 0.063±0.003 | 0.06±0.005 | 0.063±0.003 | 0.056±0.008 | 0.068±0.005 | 0.070±0.0046 | 0.07±0.005 |
| IVSs (cm) | 0.11±0.005 | 0.11±0.003 | 0.11±0.003 | 0.12±0.003 | 0.10±0.003 | 0.10±0.004 | 0.12±0.005 | 0.09±0.003** |
| LVDs (cm) | 0.23±0.010 | 0.22±0.013 | 0.21±0.006 | 0.22±0.012 | 0.24±0.025 | 0.27±0.013 | 0.21±0.013 | 0.31±0.008** |
| LVPWs (cm) | 0.11±0.003 | 0.11±0.011 | 0.12±0.003 | 0.11±0.003 | 0.11±0.003 | 0.11±0.008 | 0.12±0.005 | 0.08±0.003* |
| LVED vol (ml) | 0.13±0.008 | 0.16±0.016 | 0.14±0.012 | 0.15±0.017 | 0.15±0.024 | 0.17±0.014 | 0.14±0.011 | 0.19±0.008 |
| LVES vol (ml) | 0.033±0.003 | 0.033±0.003 | 0.026±0.003 | 0.026±0.006 | 0.036±0.012 | 0.05±0.005 | 0.026±0.003 | 0.08±0.005*** |
| EF (%) | 75.59±2.25 | 80.80±1.24 | 82.51±0.68 | 81.52±1.129 | 76.02±3.425 | 70.51±4.172 | 81.29±0.101 | 57.69±0.92**** |
| FS (%) | 38.74±1.948 | 43.58±1.214 | 45.29±0.74 | 44.31±1.16 | 39.28±2.83 | 34.85±3.18 | 43.98±0.10 | 25.92±0.54**** |
| LV eject vol (ml) | 0.10±0.011 | 0.12±0.021 | 0.12±0.015 | 0.13±0.026 | 0.11±0.021 | 0.12±0.026 | 0.11±0.015 | 0.11±0.010 |
| h/r | 0.33±0.008 | 0.32±0.022 | 0.33±0.021 | 0.33±0.032 | 0.31±0.019 | 0.31±0.019 | 0.37±0.02 | 0.31±0.001* |

Values are means ± SEM. Statistics: one-way ANOVA followed by Tukey's multiple comparison test.

IVS inter ventricular septum, LVD left ventricular diameter, LVPW left ventricular posterior wall; LVED left ventricular end diastolic, LVES left ventricular end systolic, EF ejection fraction, FS fractional shortening, s systole, d diastole.

*$p ≤ 0.05$, **$p ≤ 0.01$, ***$p ≤ 0.001$, and ***$p ≤ 0.0001$ between WT mice and WT mice transduced with AAV-HDAC6.

benefit clinical development for cardiac diseases[117]. ERK1/2 inhibitors may also be promising therapies, as treatment of lamin mutant mice with ERK1/2 signaling inhibitors can reduce cardiac and skeletal phenotypes[3–6]; this may be attributed at least in part to the effect on MRTF-A/SRF signaling. Although the effects of the long-term use of the aforementioned compounds are currently unknown, these findings encourage further approaches to target impaired MRTF-A/SRF signaling and thereby ameliorate devastating cardiac diseases associated with *LMNA* mutations. Overall, our work provides an adequate basis for testing these therapeutic approaches for patients with cardiomyopathy caused by mutations in *LMNA*.

## Methods

### Ethical statement

This research complies with all relevant ethical regulations for the boards/committees and institutions that approved the study protocols. Sections of explanted hearts from human subjects with *LMNA* mutations were obtained without identifiers from Myobank-AFM de l'Institut de Myologie. Myobank-AFM received approval from the French Ministry of Health and from the Committee for Protection of Patients to share tissues and cells of human origin for scientific purposes, ensuring the donors' anonymity, respect of their will, and consent according to the legislation. Animal experimentation was performed in agreement with the guidelines from Directive 2010/63/EU of the European Parliament regarding the protection of animals used for scientific purposes. All animal experiments were approved by the French Ministry of Higher Education and Research at the Center for Research in Myology for the care and use of experimental animals.

### C2C12 cell culture and reagents

The generation of C2-WT and C2-H222P cells was described previously[40]. Cells were cultured in proliferative Dulbecco's modified Eagle's medium (DMEM) supplemented with 10% fetal bovine serum (Life Technologies) and 0.1% gentamycin (Invitrogen). Transient transfection experiments were performed using Lipofectamine 2000 (Invitrogen) according to the manufacturer's instructions. Briefly, cells seeded at $3 × 10^5$ cells per well in six-well plates were transfected for 24 h. Plasmids encoding cofilin-1 T25A, T25D, and V20A were previously described in7. Scrambled siRNA and siRNAs silencing PDXP, SSH1, PP2B, and Atat1 were purchased from Santa Cruz Biotechnology. Working concentrations of 50 µM selumetinib were prepared from stocks diluted in DMSO. Cells were incubated with selumetinib for 15 h. Untransformed C2C12 cells (ATCC, #CRL-1772) were seeded at $3 × 10^5$ cells per well in six-well plates and were transfected using Lipofectamine 3000 (Invitrogen) with 1 µg of plasmids encoding MRTF-A (#19838, Addgene), MRTFΔ100 (#27176, Addgene) or SRF-VP16 for 48 h. SRF-VP16 plasmid was kindly provided by Dr. Sotiropoulos (Université de Paris).

### Human induced pluripotent stem cell culture and differentiation

The hiPSC carrying *LMNA* p.H222P, p.S143P, and p.R190W mutations lines were generated from patients' primary cells (Table S7) by Sendai virus infection and characterized as previously described[118]. A signed informed consent was obtained from all the individuals participating and the study was approved by the review boards of Humanitas Research Hospital (ID:1215), and Tampere Pirkanmaa Hospital District (R08070) to establish, culture, and differentiate hiPSC lines. The hiPSC lines were cultured in mTeSR1 medium (Stemcell Technologies) supplemented with 5 µM StemMACS Y27632 (Miltenyi Biotec) at 37 °C and 5% CO₂ on Matrigel hESC-Qualified Matrix (Corning Life Sciences), and the medium was refreshed daily (without Y27632). Upon reaching ~80% confluence, the cells were passaged in clumps by scraping with a pipette tip. After at least two passages, cells were differentiated using a 2D monolayer differentiation protocol and maintained in a 5% CO₂/air

**Table 3 | Echocardiographic data for *Lmna*^p.H222P/H222P (H222P) mice treated with tubastatin A or DMSO at 2, 2.5, and 3 months of age**

| | H222P DMSO | H222P Tubastatin | H222P DMSO | H222P Tubastatin | H222P DMSO | H222P Tubastatin |
|---|---|---|---|---|---|---|
| Age | 2 months | 2 months | 2.5 months | 2.5 months | 3 months | 3 months |
| *n* | 5 | 6 | 5 | 6 | 5 | 6 |
| Time (ms) | 99.83 ± 1.65 | 100.4 ± 1.75 | 101.0 ± 1.56 | 100.4 ± 1.72 | 102.3 ± 2.09 | 100.7 ± 1.35 |
| Heart rate (bpm) | 601.7 ± 9.96 | 598.3 ± 10.14 | 594.5 ± 9.31 | 598.7 ± 10.25 | 587.4 ± 11.93 | 596.4 ± 7.94 |
| Cardiac output (l/min) | 0.06 ± 0.007 | 0.07 ± 0.002 | 0.07 ± 0.002 | 0.06 ± 0.003 | 0.06 ± 0.005 | 0.06 ± 0.003 |
| IVSd (cm) | 0.07 ± 0.002 | 0.07 ± 0.02 | 0.07 ± 0.001 | 0.07 ± 0.001 | 0.07 ± 0.002 | 0.07 ± 0.001 |
| LVDd (cm) | 0.37 ± 0.01 | 0.37 ± 0.004 | 0.39 ± 0.002 | 0.37 ± 0.004 | 0.39 ± 0.009 | 0.37 ± 0.003 |
| LVPWd (cm) | 0.07 ± 0.003 | 0.07 ± 0.003 | 0.06 ± 0.003 | 0.07 ± 0.003 | 0.07 ± 0.004 | 0.06 ± 0.002 |
| IVSs (cm) | 0.12 ± 0.004 | 0.12 ± 0.002 | 0.11 ± 0.002 | 0.11 ± 0.002 | 0.11 ± 0.004 | 0.11 ± 0.003 |
| LVDs (cm) | 0.21 ± 0.006 | 0.21 ± 0.004 | 0.24 ± 0.004 | 0.19 ± 0.004**** | 0.26 ± 0.008 | 0.22 ± 0.005**** |
| LVPWs (cm) | 0.0 ± 0.004 | 0.11 ± 0.003 | 0.09 ± 0.004 | 0.11 ± 0.003 | 0.08 ± 0.002 | 0.10 ± 0.005* |
| LVED vol (ml) | 0.13 ± 0.11 | 0.13 ± 0.004 | 0.15 ± 0.003 | 0.12 ± 0.003 | 0.16 ± 0.01 | 0.12 ± 0.005 |
| LVES vol (ml) | 0.02 ± 0.003 | 0.02 ± 0.002 | 0.04 ± 0.002 | 0.02 ± 0.001** | 0.05 ± 0.005 | 0.03 ± 0.003** |
| EF (%) | 82.9 ± 0.46 | 82.45 ± 0.52 | 75.33 ± 0.39 | 82.90 ± 0.58*** | 70.18 ± 2.01 | 78.81 ± 1.11**** |
| FS (%) | 45.65 ± 1.06 | 45.25 ± 0.55 | 38.43 ± 0.33 | 45.68 ± 0.63**** | 34.42 ± 1.51 | 41.60 ± 1.03**** |
| LV eject vol (ml) | 0.10 ± 0.009 | 0.11 ± 0.002 | 0.11 ± 0.002 | 0.10 ± 0,003 | 0.11 ± 0.007 | 0.10 ± 0.003 |
| h/r | 0.37 ± 0.006 | 0.35 ± 0.008 | 0.34 ± 0.007 | 0.38 ± 0.005* | 0.35 ± 0.008 | 0.36 ± 0.004 |

Statistics: one-way ANOVA followed by Tukey's multiple comparison test. Values are means ± SEM.

*IVS* inter ventricular septum, *LVD* left ventricular diameter, *LVPW* left ventricular posterior wall, *LVED* left ventricular end diastolic, *LVES* left ventricular end systolic, *EF* ejection fraction, *FS* fractional shortening, s systole, d diastole.

*$p \leq 0.05$, **$p \leq 0.01$, ***$p \leq 0.001$ and ****$p \leq 0.0001$ between age-matched tubastatin A and DMSO treated mice.

environment. Briefly, the cells were detached using ReLeSR and replated onto Matrigel-coated 12-well plates in mTeSR1 medium. At 85% cell confluence, the cells were treated for 2 days with 6 μM CHIR99021 (Abcam) in RPMI + B27 supplement without insulin. On day 2, the medium was replaced with RPMI + B-27 medium without insulin or CHIR99021. On days 3–5, the cells were treated with 5 μM IWR1 (Sigma). On days 5–6, the cells were removed from the medium containing IWR-1 and placed in RPMI + B27 medium without insulin. From day 7 onwards, the cells were cultured in RPMI + B27 medium with insulin, and the medium was changed every two days. When replating for future use, iPSC-CMs were dissociated with 0.25% trypsin-EDTA into a single-cell suspension and seeded onto Matrigel-coated plates or micropatterned dished (4D cell). Then, the cells were washed with phosphate-buffered saline (PBS), lysed with protein extraction buffer (Cell Signaling), and frozen at −80 °C. iPSC-CMs were treated with tubastatin A (SelleckChem) dissolved in DMSO at 3 μM for 24 h.

### Mouse strains, husbandry, and treatments
Wild-type C57/Bl6, *Lmna*^p.H222P/H222P. [47] and *Atat1*^−/− male mice[54] were fed standard chow and water ad libitum and housed in a disease-free barrier facility under a 12 h/12 h light/dark cycle, controlled temperature of 22 °C, and 55% humidity. All animal experiments were approved by the French Ministry of Higher Education and Research at the Center for Research in Myology for the Care and Use of Experimental Animals. The animal experiments were performed according to the guidelines from Directive 2010/63/EU of the European Parliament regarding the protection of animals used for scientific purposes. Tubastatin A (SelleckChem) dissolved in DMSO was delivered at a dose of 10 mg/kg daily to *Lmna*^p.H222P/H222P mice. Tubastatin A and placebo (DMSO) were administered to mice by intraperitoneal injection beginning at 2 months of age and continuing for one month.

### Isolation of mouse cardiomyocytes
WT and *Lmna*^p.H222P/H222P mice were anesthetized via intraperitoneal injection of sodium pentobarbital (50 mg/kg), and ventricular cardiomyocytes were isolated as described previously[119]. In brief, mouse hearts were removed and placed in ice-cold oxygenated Tyrode's solution. The aorta was cannulated above the aortic valve and perfused with Tyrode's solution containing 0.1 mmol/l EGTA for 2 min. An enzyme mixture containing 1 g/liter collagenase Type II (Worthington Biochemical Corporation) in Tyrode's solution supplemented with 0.1 mmol/l CaCl₂ was then perfused until the aortic valve was digested. The heart was removed, cut into small pieces, and gently shaken in enzyme solution supplemented with 2 g/liter BSA for 2 min at 37 °C to disperse individual myocytes. The myocytes were then filtered through a 250-μm nylon mesh and centrifuged at $200 \times g$ for 2 min. A pellet containing myocytes was resuspended in Tyrode's solution supplemented with 0.5 mmol/l CaCl₂ and 2 g/liter BSA and then recentrifuged. The final pellet was resuspended in a storage buffer comprising Tyrode's solution supplemented with 1 mmol/l CaCl₂. Isolated cardiomyocytes were used immediately, either protein were extracted for western blotting or cells were fixed for immunostaining.

### Human heart tissue
Sections of explanted hearts from human subjects with *LMNA* mutations were obtained without identifiers from Myobank-AFM de l'Institut de Myologie. Myobank-AFM received approval from the French Ministry of Health and from the Committee for the Protection of Patients to share tissues and cells of human origin for scientific purposes, ensuring the donors' anonymity, respect of their volition, and consent according to the legislation. The subjects were a 23-year-old man with cardiomyopathy associated with muscular dystrophy and *LMNA* p.delK261 mutation, a 53-year-old man with cardiomyopathy and *LMNA* p.E33D mutation, and a 47-year-old woman with cardiomyopathy and *LMNA* p.R60G mutation. Control human heart samples were obtained from a 57-year-old man with an intracranial bleed, a 15-year-old woman who died of a drug overdose, and a 46-year-old man who died from end-stage liver disease.

### Protein extraction and immunoblotting
Total proteins were isolated from cultured cells, cardiac sections, or isolated cardiomyocytes with extraction buffer (Cell Signaling) containing protease inhibitors (25 mg/ml aprotinin, 10 mg/ml leupeptin,

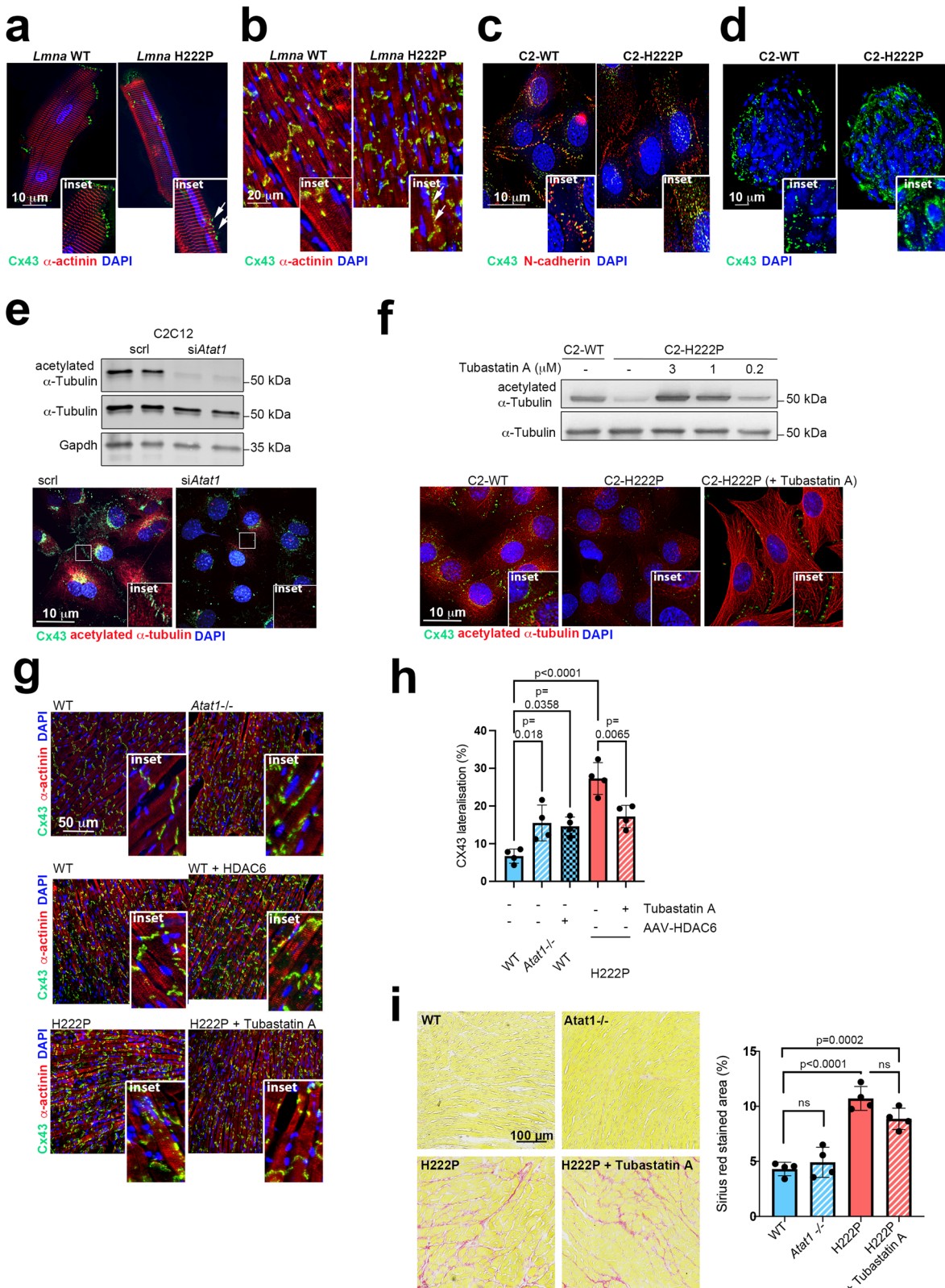

1 mM 4-[2-aminoethyl]-benzene sulfonyl fluoride hydrochloride and 2 mM Na$_3$VO$_4$. The lysates were sonicated (3 pulses of 10 s at 30% amplitude). Cytosolic and nuclear fractions were prepared using NE-PER Nuclear and Cytosolic Extraction Reagents (Thermo Fisher Scientific) according to the manufacturer's instructions. Sample protein content was determined by the bicinchoninic acid protein assay (Thermo Fisher Scientific). Protein extracts (10–20 μg) were analyzed by SDS–PAGE on a 10% gel and transferred onto nitrocellulose membranes (Invitrogen). After washing with Tris-buffered saline containing 1% Tween 20 (TBS-T), the membranes were blocked with 5% bovine serum albumin (BSA) in TBS-T for 1 h at room temperature and then incubated with the appropriate antibody overnight at 4 °C. Membranes were incubated with fluorescent-conjugated anti-mouse or anti-rabbit secondary antibodies (BioRad) 1 h at RT. Antibody

**Fig. 6 | α-Tubulin acetylation mediates Cx43 localization in cellular and mouse models expressing cardiomyopathy-causing mutant A-type lamins.**
**a**, **b** Immunofluorescence staining of Cx43 and α-actinin in **a** isolated adult cardiomyocytes and **b** heart cross-sections from 3-month-old male WT mice and *Lmna*^p.H222P/H222P (H222P) mice. The arrows indicate the lateralization of Cx43.
**c** Immunofluorescence staining of Cx43 and N-cadherin in C2-WT and C2-H222P cells. **d** Immunofluorescence staining of Cx43 in C2-WT and C2-H222P cells organized in 3D spheroids. **e**, **f** Top: immunoblot showing expression of acetylated and total α-tubulin in **e** C2C12 cells treated or not with siRNA targeting *Atat1* and **f** in C2-H222P cells treated or not with different doses of tubastatin A. GAPDH was used as a loading control. **e**, **f** Bottom: representative immunofluorescence staining of Cx43 and acetylated α-tubulin in **e** C2-H222P cells treated or not with a siRNA targeting *Atat1* and **f** in C2-H222P cells treated or not with tubastatin A. The insets show a higher magnification. **g** Immunofluorescence staining of Cx43 and α-actinin of

heart cross-sections from 6-month-old male *Atat1*^+/+ (WT) mice and *Atat1* knockout mice (*Atat1*^−/−) (top panel); from 5-month-old male WT mice transduced or not with AAV expressing HDAC6 (middle panel) and from 3-month-old male *Lmna*^p.H222P/H222P (H222P) mice treated with tubastatin A compared with mice treated with DMSO (bottom panel). **h** Bar graph showing the quantification of lateralized Cx43 staining (*n* = 4 mice per condition; mean ± SD). **i** Left: representative Sirius red staining of heart cross-sections from 6-month-old male *Atat1*^+/+ (WT) mice, *Atat1* knockout mice (*Atat1*^−/−), and 3-month-old male *Lmna*^p.H222P/H222P (H222P) mice treated or not with tubastatin A. Right: bar graph showing the quantification of Sirius red-stained area (*n* = 4 mice per condition; mean ± SD). For **a**–**g**, nuclei counterstained with DAPI are shown and a representative of three independent repeats is shown. Statistics: for **h** and **i**, one-way ANOVA followed by Tukey's multiple comparison test. Source data are provided as a Source Data file.

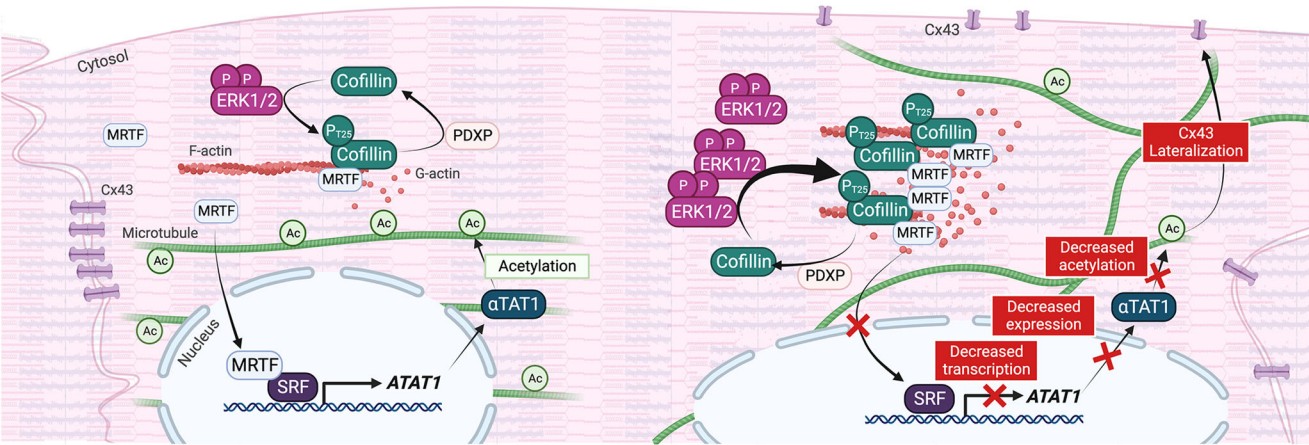

**Fig. 7 | Schematic representation of the actin-microtubule interplay mediated by phospho(T25)-cofilin-1 via MRTF-A/SRF signaling and α-tubulin acetylation in *LMNA*-cardiomyopathy.** Created with BioRender.com.

---

detection was imaged using the ChemiDoc Imaging System and ImageLab software (BioRad). Quantification was performed using FIJI/ImageJ software.

## Immunofluorescence microscopy

For immunofluorescence microscopy, C2C12 and iPS-CM cells were grown on coverslips, washed with PBS, and fixed with 4% paraformaldehyde in PBS for 10 min. The cells were permeabilized with 0.2% Triton X-100 diluted in PBS for 10 min, and nonspecific signals were blocked with 0.2% Triton X-100 and 5% BSA for 45 min. The samples were then incubated for 1 h with the primary antibody in PBS supplemented with 0.1% Triton X-100 and 1% BSA at room temperature. Cells and slides were then mounted in Vectashield mounting medium with 4',6-diamidino-2-phenylindole (DAPI) (Vector Laboratories).

Frozen cardiac tissues cut into 8 μm-thick cryosections, were fixed (15 min, 4% paraformaldehyde in PBS at room temperature), permeabilized (10 min, 0.5% Triton X-100 in PBS), and blocked (1 h, PBS with 0.3% Triton X-100, 5% BSA). The sections were incubated with primary antibodies (in PBS with 0.1% Triton X-100 and 1% BSA) overnight at 4 °C and then washed with PBS. The sections were then incubated with the appropriate secondary antibodies for 1 h before being washed with PBS, and slides were mounted in Vectashield mounting medium with DAPI (Vector Laboratories).

Immunofluorescence microscopy was performed using an Axiophot microscope (Carl Zeiss) or a 3i Marianas spinning disk confocal with a Yokogawa CSU-W1 scanning unit on an inverted Zeiss AxioObserver Z1 microscope controlled by SlideBook 6 software (Intelligent

Imaging Innovations GmbH, Göttingen, Germany). A ×63/1.4 oil objective was utilized, and images were acquired with an ORCA Flash4 sCMOS camera (Hamamatsu Photonics, Hamamatsu, Japan). Most of the images were digitally deconvolved using Autodeblur v9.1 (Autoquant) deconvolution software and processed using Adobe Photoshop 6.0 (Adobe Systems). A Nikon Ti2 microscope equipped with a motorized stage and coupled with Prime 95B Scientific CMOS (sCMOS) camera at ×100 with Oil Immersion objective was used for some of the acquisitions. The microscope was controlled by MetaMorph 7.10 software (Molecular Devices) with a pixel resolution of 0.11 μm/px at 16-bit. The images were processed by FIJI/ImageJ software.

## F/G actin ratio measurements

The ratio of F-actin to G-actin was determined using the G-actin/F-actin in vivo assay kit (Cytoskeleton) according to the manufacturer's instructions. Briefly, 2 mg of protein from cells or frozen heart tissues were homogenized in lysis and F-actin stabilization buffer and centrifuged at 350 × *g* for 5 min to remove unbroken cells. F-actin was separated from G-actin by centrifugation at 100,000 × *g* for 60 min at 37 °C. The F-actin-containing pellet was resuspended in F-actin depolymerizing buffer at a volume equivalent to the G-actin-containing supernatant volume. The resuspended F-actin pellet was kept on ice for 60 min and was gently mixed every 15 min to dissociate F-actin. Proteins in equivalent volumes (10 μl) of supernatant and pellet were separated by SDS−PAGE and subjected to immunoblot analysis using an anti-pan actin antibody supplied in the kit. F/G actin ratio was quantified using ImageJ software.

## Proximity ligation analysis

The Duolink in situ proximity ligation assay (PLA) was performed according to the manufacturer's protocol (Sigma-Aldrich). Briefly, C2-WT and C2-H222P cells were plated on glass coverslips. The cells were washed, blocked, and then incubated with anti-MRTF-A (1:25; Santa Cruz Biotechnology), anti-SRF (1:25; Santa Cruz Biotechnology), or anti-phospho(T25)-cofilin-1 (1:10; Genscript) primary antibodies overnight at 4 °C. After washing, the cells were incubated with the Duolink PLA probes MINUS and PLUS for 1 h at 37 °C. A Duolink in situ detection kit was used for ligation and amplification, and nuclei were stained with Hoechst.

## Immunoprecipitation

Cells were lysed in 0.5 ml of lysis buffer (50 mM Tris−HCl [pH 7.5], 0.15 M NaCl, 1 mM EDTA, 1% NP-40), precleared with 20 µl of washed protein G−Sepharose 4 fast-flow (GE Healthcare) and incubated with 20 µg of a specific antibody overnight at 4 °C. Next, the washed beads (30 µl) were added and incubated for 2 h at 4 °C. Pelleted beads were collected in sample buffer (0.25 M Tris−HCl [pH 7.5], 8% SDS, 40% glycerol, 20% β-mercaptoethanol) and subjected to SDS−PAGE and immunoblotting. For control reactions, we used rabbit immunoglobulin G1.

## Luciferase assays

The SRF reporter construct (3DA-luciferase: 3 CArGs upstream of the luciferase gene) was described previously[21]. The pGL4.3.4[luc2P]/SRF-RE/Hygro vector (E1350; Promega) contains a sequence of five consensuses CArG boxes in tandem separated by nucleotide spacers followed by a minimal promoter upstream of a modified firefly luciferase gene. This vector was used to generate the plasmid *ATAT1* CArGs1−4 as described in ref. [49]. The DNA constructs (50 ng) were transfected into C2C12 cells at 70% confluency together with 1.5 ng of a Renilla luciferase reporter plasmid, which was used to normalize for transfection efficiency, using Lipofectamine 2000. Forty-eight hours after transfection, the activities of firefly and Renilla luciferase were measured using a dual-luciferase reporter assay system (Promega) according to the manufacturer's instructions on a TECAN SPARK 10 M microplate reader. SparkControl software was used to quantify luminescence. Each experiment was performed in triplicate.

## RNA isolation and reverse-transcription qPCR

Total RNA was extracted from C2C12 or mouse heart using RNeasy isolation kit (Qiagen) according to the manufacturer's instructions. The adequacy and integrity of the extracted RNA were determined with the 2100 Bioanalyzer system (Agilent) according to the manufacturer's instructions. cDNA was synthesized using the SuperScript III first-strand synthesis system according to the manufacturer's instructions (Invitrogen). Real-time qPCR was performed with SYBR Green I Master mix (Roche) on the LightCycler 480 instrument and analyzed with LightCycler 480 software (Roche). The primers utilized in this study are listed in Table S8.

## Adeno-associated virus (AAV) delivery

AAV delivery was performed in three-month-old mice by retro-orbital injection using a 30-G needle, at the dose of $5 \times 10^{13}$ viral genomes/kg in 100 µl. Phosphate-buffered saline; PBS 1X was used as a placebo. Mice were anesthetized with an intraperitoneal injection of xylazine (10 mg/kg)/ketamine (100 mg/kg) cocktail and placed on a heating pad at 28 °C during the intervention. Wild-type males were injected with AAV-rh10-Cfl1, AAV-rh10-Cfl1(p.T25A) at 3 months of age. AAV vectors of serotype rh10 (AAV-rh10-Cfl1, AAV-rh10-Cfl1(p.T25A)), carrying sequence of the wild-type cofilin-1 gene or cofilin-1 c.73A>G under the control of the cytomegalovirus immediate/early promoter was prepared by the triple transfection method in HEK293T cells (ATCC #CRL-1573) as previously described[7,120].

## Echocardiography

Mice were anesthetized with 0.75% isoflurane in $O_2$ and placed on a heating pad (28 °C). Transthoracic echocardiography was performed using a Vivid 7 Dimension/Vivid7 PRO ultrasound with an 11 MHz transducer applied to the chest wall. Cardiac ventricular dimensions and fractional shortening were measured in 2D mode and M-mode three times for the number of animals indicated. A 'blinded' echocardiographer, unaware of the genotype and the treatment, performed the examinations.

## Electrocardiography

Electrocardiograms were recorded from mice using the non-invasive ecgTUNNEL (Emka Technologies) with minimal filtering. Waveforms were recorded using Iox Software and intervals were measured manually with ECG Auto software. The electrocardiographer was blinded to the mouse genotype.

## Microarray processing

We used GeneChip Mouse Genome 430 2.0 Arrays (Affymetrix), which contained 698,000 probes that covered 35,240 transcripts from the RefSeq database. RNA was extracted as described above and cDNA synthesis, cRNA synthesis, and labeling were performed as described in the Affymetrix GeneChip Expression Analysis Technical Manual. Hybridization, washing, staining, and scanning of the arrays were performed at the GeneChip Core Facility of Cochin Hospital (GENOM'IC). Image files were obtained with Affymetrix GeneChip software and subjected to robust multichip analysis using Affymetrix microarray ".cel" image files and GeneTraffic 3.0 software (Stratagene). The robust multichip analysis involved 3 steps: background correction, quantile normalization, and probe set summary. Genes were identified as differentially expressed if they showed at least a 2-fold difference in expression independent of the absolute signal intensity and met a false discovery rate threshold of $q < 0.05$ as determined by a 2-tailed Student's $t$-test. Data generated have been deposited in the Gene Expression Omnibus database under accession code GSE218891

## Analysis of functional groups of genes

Gene expression changes related to functional groups were analyzed using the class score method in ermineJ (version 2.1.12; http://www.bioinformatics.ubc.ca/ermineJ/) to provide statistical confidence values for the functional groupings. The algorithm utilizes the log-transformed 2-tailed Student's $t$-test $P$ values of genes that are members of a single Gene Ontology class as input and estimates the probability of the set of $q$ values occurring by chance. Significant Gene Ontology terms were identified based on a false discovery rate of 0.05. For automated functional annotation and the classification of genes of interest based on Gene Ontology terms, we used the DAVID database (http://david.abcc.ncifcrf.gov/).

## Three-dimensional C2C12 cultivation (spheroid model)

C2C12 cells were dissociated and resuspended in a culture medium at a concentration of 125,000 or 62,500 cells per ml. In total, 200 µl of the cell suspension was plated into each well of 96-well polystyrene round bottom plastic plates (Nunc) coated with a 1% agarose/water solution as previously described[121]. The cells were cultured at 37 °C in a 5% $CO_2$ atmosphere for 24 h and then fixed with 4% paraformaldehyde for immunocytochemistry.

## Histological analysis

Frozen hearts were cut into 8-µm-thick sections and stained with Sirius red. Briefly, heart sections were fixed in 4% formaldehyde for 10 min, rinsed in EtOH 100%, dried 20 min, and stained in 0.3% Sirius red solution for 1 h. Then sections were put in acetic acid 0.5% for 5 min twice, in EtOH 100% for 5 min three times, and finally in xylene for

10 min twice. Collagen fibrils were detected in red whereas cytoplasm remained yellow.

## MuscleMotion analysis
MUSCLEMOTION[122] ImageJ macro was used as described in the user's guide to quantify cardiomyocytes derived from iPS contractions.

## Antibodies
The primary antibodies used were anti-cofilin1 (Cell Signaling), custom-made anti-phospho(T25)-cofilin-1 (GenScript, dilution 1:50), anti-phospho(Ser3)-cofilin-1 (Cell Signaling), anti-phosphoERK1/2 antibody (Cell Signaling), anti-ERK1/2 (Cell Signaling), anti-MRTF-A (Santa Cruz Biotechnology), anti-Cx43 (Abcam), anti-α-tubulin (Abcam), anti-acetylated α-tubulin (Sigma), anti-Flag (Sigma), anti-GAPDH (Abcam), anti-N-cadherin (Cell Signaling), anti-Cx43 (Abcam), anti-actin (Cytoskeleton), anti-cardiac troponin T (Cell Signaling), anti-HDAC6 (Cell signaling), and anti-α-actinin sarcomeric (Sigma). The secondary antibodies for immunofluorescence were Alexa Fluor 488-conjugated goat anti-rabbit IgG, Alexa Fluor 568-conjugated goat anti-mouse IgG, and Alexa Fluor 488-conjugated donkey anti-goat IgG (Life Technologies). For immunoblotting, fluorescent-conjugated StarBright Blue 520 anti-mouse or StarBright Blue 700 anti-rabbit secondary antibodies (BioRad) were used. All the antibodies were used at dilutions recommended by the manufacturer.

## Statistics
Statistical analyses were performed using GraphPad Prism software. Statistical significance between groups of mice subjected to echocardiography was analyzed with a corrected parametric test (Welch's $t$-test), with $P < 0.05$ being considered significant. To validate the results of the echocardiographic analyses, we performed a nonparametric test (Wilcoxon–Mann–Whitney test). For all other experiments, either a two-tailed Student's $t$-test or one-way ANOVA was used, with $P < 0.05$ being considered significant. Values are presented as the mean ± standard deviation (SD), and the sample sizes are indicated in the figure and table legends.

## Reporting summary
Further information on research design is available in the Nature Portfolio Reporting Summary linked to this article.

# Data availability
All data generated or analyzed in this study are provided in the Source Data file/Supplementary Information file. The Affymetrix microarray data generated in this study have been deposited in the Gene Expression Omnibus database under accession code GSE218891. Source data are provided with this paper.

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

## Acknowledgements

The work of A.M. was supported by the Institut National de la Santé et de la Recherche Médicale, Sorbonne Université, and the Association Française contre les Myopathies, the French National Research Agency (ANR, award ANR-19-CE17-0013), MyoPharm and a grant (PER-SU) provided by Sorbonne Université. C.J. received support from ANR-10-IDEX-0001-02, the LabEx CellNScale ANR-11-LBX-0038, the French National Research Agency (ANR, award ANR-17-CE13-0021), and the Fondation pour la Recherche Medicale (FRM, grant DEQ20170336756). P.T. was supported by research funding provided by the Academy of Finland. We thank the genomic facility at Institut Cochin, Paris, for Affymetrix data acquisition. We thank Dr. Ben Yaou (Institute of Myology Paris) and Dr. Eschenhagen (University Medical Center Hamburg-Eppendorf) for providing the iPSCs carrying the LMNA p.H222P mutation. We thank Dr. Dal Ferro (University of Trieste) for providing the iPSCs carrying the LMNA p.R190W mutation. We thank Dr. Alonso (Universidad Autónoma de Madrid) for providing the ATAT1 CArGs 1–4 construct and Dr. Sadoul (Université Grenoble Alpes) for providing the ATAT knockout mouse strain. We thank Dr. Sotiropoulos (Université de Paris) for providing the SRF-VP16 plasmid. We thank Zoheir Guesmia for his help with imaging and quantification. Parts of Figs. 3 and 4 were drawn using pictures from Servier Medical Art. Servier Medical Art by Servier is licensed under a Creative Commons Attribution 3.0 Unported License. Fig. 7 was created thanks to BioRender.com.

## Author contributions

Conceptualization, A.M.; investigation, C.L.D., M.C., C.M., C.P., B.C., D.C., and D.A.; *ATAT1* mouse model, M.M.M. and C.J.; iPS cells, C.J., L.V., T.H., K.A.-S., S.C., E.D.P., P.T., and J.-S.H.; cardiac function assessment, N.M. and A.M.; writing—original draft, A.M.; writing—review and editing, C.L.D., P.T., M.C., and C.J.; funding acquisition, A.M.; supervision, A.M.

## Competing interests

The authors declare no competing interests.

## Additional information

¹Centre de recherche en Myologie, U974 SU-INSERM, 75013 Paris, France. ²Institut Curie, Université PSL, CNRS UMR3348, 91401 Orsay, France. ³Université Paris-Saclay, CNRS UMR3348, 91401 Orsay, France. ⁴Université de Paris, Paris Cardiovascular Research Center PARCC, INSERM, 75015 Paris, France. ⁵Institute of Biomedicine and FICAN West Cancer Centre, University of Turku, 20520 Turku, Finland. ⁶Heart and Lung Centre, Helsinki University Central Hospital, University of Helsinki, 00290 Helsinki, Finland. ⁷Faculty of Medicine and Health Technology and BioMediTech Institute, Tampere University, 33520 Tampere, Finland. ⁸Institute of Genetic and Biomedical Research (IRGB), UOS of Milan, National Research Council of Italy, Milan, Italy. ⁹Humanitas Clinical and Research Center-IRCCS, 20089 Rozzano (MI), Italy. ¹⁰Sorbonne Université, INSERM, UMS28 Phénotypage du petit animal, Paris 75013, France. ¹¹Fundação Oswaldo Cruz, Instituto Oswaldo Cruz, Laboratório de Biologia Estrutural, Rio de Janeiro, Brasil. ¹²Department of Pathology, Turku University Hospital, 20520 Turku, Finland. ¹³These authors contributed equally: Caroline Le Dour, Maria Chatzifrangkeskou, Coline Macquart. ✉e-mail: a.muchir@institut-myologie.org

