## [Peer Review File · Nature Communications]

Actin-microtubule cytoskeletal interplay mediated by MRTF-A/SRF signaling promotes dilated cardiomyopathy caused by LMNA mutationsREVIEWER COMMENTS

Reviewer #1 (Remarks to the Author):

In this study, the authors performed detail evaluation of the potential role of MRTF-A/SRF signaling pathway as the pathogenic mechanisms of LMNA mutation cardiomyopathy. They used a skeletal muscle cell line (C2C12) and human iPSC line and mouse model carrying LMNA mutation to show the involvement of actin-microtubule interaction mediated by MRTF-A/SRF signaling pathway. They further demonstrated that abnormal α -tubulin acetylation can affect the Cx34 location and thus the cardiac function. Furthermore, treatment with HDAC6 inhibitor restored the α -tubulin acetylation and trafficking of Cx43. Overall, the results are of potential and provide important insights on the mechanism of LMNA cardiomyopathy. However, there are several major concerns need to be addressed.

1. There are many type of LMNA mutations, but the majority of those caused dilated cardiomyopathy only involved the cardiomyocytes rather than the skeletal muscle. Therefore, there are major concerns on the use of C2C12 cell lines as a model in this study. Indeed, the authors have generated several human iPSC derived cardiomyocytes line in which some of the experiments should be confirmed with those human iPSC- cardiomyocytes, such as the treatment effects of tubastatin as they have already showed the reduction in the α -tubulin acetylation in those human cells.
2. There are lack of any functional assay in the in-vitro experimental. Are there any changes in the contractile function in the mouse and human iPSC cardiomyocytes carrying the mutation before and after treatment with tubastatin.
3. Clinically, one of the most important clinical phenotype of LMNA cardiomyopathy is conduction defects with heart block before the occurrence of heart failure. As the authors nicely demonstrated the changes in the trafficking of Cx43 before and after tubastatin, are there any changes in ECG and occurrence of proarrhythmic phenotypes in the mouse and human iPSC model of LMNA cardiomyopathy.
4. One of the other important phenotype of LMNA cardiomyopathy is the aberrant nuclear morphology and specific disruptions in peripheral chromatin. Are there any changes in the nuclear morphology in those mutant cells as well as in the mouse cardiomyocytes?
5. The authors included several human iPSC lines with LMNA mutation in this study, were those patients have both skeletal and cardiac phenotypes? Their clinical information should be included in the supplemental data.

Reviewer #2 (Remarks to the Author):

In the manuscript by LeDour et al, the authors try to decipher a pathway connecting LMNA mutations in cardiomyopathy to cofilin/SRF/MRTF-A signalling and further downstream to altered tubulin acetylation and Connexin 43 mis-localization via transcriptional regulation of the ATAT1 acetyltransferase.

Overall, the manuscript is well written and some of individual parts of the manuscript, particular the part on acetylation and connexin regulation are experimentally sound and very interesting (Fig. 5 and 6). However, data for several other molecular connections made in this manuscript are not always convincing. In particular, the regulation of the ATAT1 gene by SRF/MRTF which is central to their model is less robust and lacks more experimental evidence as detailed below. Furthermore, there are discrepancies between their experimental cell line C2 and the human iPSC derived cell lines from patients which suggest that their model might not have such a translational impact.

Thus, in summary some of the experimental findings are interesting and also novel and would be interesting to be published in Nature Communications. However, the individual parts are not well connected and data for some parts of their signalling model are not convincing. Thus, I would support rejection of the manuscript as it stands but encourage the authors for a major revision including several new experiments and further quantification of existing data.

Specific points:

1.) Figure 1: The data obtained in C2 cells in Figure 1 are convincing in contrast to the data presented in human iPSCs carrying a LMNA mutation (Figure 1E). Here, a cytoplasmic enrichment of MRTF-A is far from convincing and the quantification is only showing single cells. Therefore, it is unclear how relevant aberrant MRTF-A localization really is for patient-derived cells. In general, this way of quantifying immunofluorescence data (single intensity blots; Fig. 1E) is done by the authors at many positions in many Figures throughout the manuscript. This quantification does not provide any N numbers, average values, SD and statistical testing. Therefore, it is difficult to judge the robustness of data. Such a robust data analysis should be performed for all immunofluorescence data.

2.) The different cofilin phosphorylation sites should be better introduced in the introduction since most people are familiar with the Ser3 site rather than the T25 site (which is very interesting). The Ser3 P site should also be analyzed in their system (Fig. 2) since one might expect reciprocal regulation to the T25 site. In Fig. 2A total cofilin levels are missing and should be provided. Fig. 2B, D, F once again lack proper quantification (see point 1). In the MRTF/cofilin IP experiment (Fig. 2h) no levels for IB: MRTF-A are provided.

3.) My major concern is with the regulation of ATAT1 gene by SRF/MRTF-A (Fig. 4). Here, the authors only provide luciferase assays which are highly artificial and are in my point of view not sufficient to show a clear regulation of ATAT1 by SRF/MRTF-A that can support convincingly the author's model. Also, the authors refer to a previous paper by Fernandez-Barrera (JCB) for the ATAT1 regulation by SRF. But also in this paper only luciferase assays were performed. The authors should use SRF/MRTF loss- (mutant cells or siRNA as done for cofilin phosphatases) and gain-of-function (e.g. SRF-VP16 overexpression) to show changes in ATAT1 endogenous mRNA regulation. Also, CHIP should be performed in a relevant cell line to clearly document ATAT1 promoter occupancy by SRF. In Fig. 4F there are 5 samples for LMNA patients in the blot, however only four are quantified. How come this discrepancy?

Minor point: on page 7 line 205 the authors refer to previous findings but it is not obvious what they are relating to (Figure, paper?).

Reviewer #3 (Remarks to the Author):

The manuscript by Le Dour et al. describes the cellular consequences of mutations in the lamin A/C gene which is a cause of dilated cardiomyopathy. Both effects on the actin cytoskeleton as well as on the microtubules were already observed before and here both observations are connected. Phosphorylated cofilin 1 (by ERK1/2) binds to myocardin-related transcription factor 1 (MRTF-A) thus preventing the stimulation of serum response factor (SRF) in the nucleus. This leads to the decreased alpha-tubulin acetylation by a reduced expression of alpha-tubulin acetyltransferase A. Increasing alpha tubulin acetylation by inhibiting HDAC6 has an interesting therapeutic effect.

Major remarks

- While after reading this manuscript the impression exists that the observation of defects in acetylation of microtubules and the connection with the mislocalization of Cx43 are a completely new observation, a (partially) similar conclusion was formulated in the 2019 HMG paper of the same research group ("...microtubule cytoskeleton alteration and decreased acetylation of α -tubulin lead to remodeling of Cx43 in LMNA cardiomyopathy, which alters the correct communication between cardiomyocytes,..."). This paper is mainly cited in the current manuscript to make the point that mutations in LMNA are associated with microtubule network alterations. In fact, much more evidence is present in the HMG manuscript linking already acetylation of microtubules with the observed phenotypes. Moreover, the potential therapeutic role of paclitaxel is not mentioned in the current manuscript. As a consequence, a much better integration/discussion of the results obtained and published before is crucial.

- While the title focuses on the actin-microtubule interplay, the consistently lower alpha-tubulin acetylation (published already before in the HMG paper) but mainly the therapeutic potential of

modulating this tubulin acetylation by inhibiting HDAC6 deserves much more attention. At present, this therapeutic effect is only mentioned at the end of the abstract without even linking it to HDAC6. Moreover, it seems to be crucial to find out how the reduction in alpha-tubulin acetylation is mechanistically linked to a loss of Cx43 localization at intercalated discs.

Minor remarks

- While it is clearly shown that tubastatin A has a positive effect on acetylation of alpha-tubulin, which subsequently also has a positive effect on the Cx43 localization, it needs to be excluded that at least part of this effect is due to a non-specific effect of tubastatin A on other HDACs. This can be easily proven by determining the (absence of an) effect of tubastatin A on histone acetylation.
- Tubastatin A (not tubastatin) should be introduced much better in the abstract.
- The acetylation level of alpha tubulin has many more effects on microtubule than just stabilizing them. This should be discussed in more detail.
- Is the expression level of HDAC6 affected by the presence of LMNA mutations?
- A broader discussion on the therapeutic possibilities and limitations of HDAC6 inhibition should be included in the discussion. At present, only a reference to a review on HDAC6 in cancer is included. This is insufficient, especially given the fact that tubastatin A is not used clinically to treat cancer. Moreover, tubastatin A belongs to a chemical class of molecules which could compromise its clinical use.

Actin-microtubule cytoskeletal interplay mediated by MRTF-A/SRF signaling promotes dilated cardiomyopathy caused by LMNA mutation

Response to reviewers' comments

We deeply appreciate the valuable and constructive review of our paper. We have incorporated the reviewers' suggestions in the revised version, which have significantly improved our paper. A point-by-point response to the reviewers' comments follows.

Reviewer #1 (Remarks to the Author):

In this study, the authors performed detail evaluation of the potential role of MRTF-A/SRF signaling pathway as the pathogenic mechanisms of LMNA mutation cardiomyopathy. They used a skeletal muscle cell line (C2C12) and human iPSC line and mouse model carrying LMNA mutation to show the involvement of actin-microtubule interaction mediated by MRTF-A/SRF signaling pathway. They further demonstrated that abnormal α -tubulin acetylation can affect the Cx34 location and thus the cardiac function. Furthermore, treatment with HDAC6 inhibitor restored the α -tubulin acetylation and trafficking of Cx43. Overall, the results are of potential and provide important insights on the mechanism of LMNA cardiomyopathy. However, there are several major concerns need to be addressed.

1. There are many type of LMNA mutations, but the majority of those caused dilated cardiomyopathy only involved the cardiomyocytes rather than the skeletal muscle. Therefore, there are major concerns on the use of C2C12 cell lines as a model in this study. Indeed, the authors have generated several human iPSC derived cardiomyocytes line in which some of the experiments should be confirmed with those human iPSC-cardiomyocytes, such as the treatment effects of tubastatin as they have already showed the reduction in the α -tubulin acetylation in those human cells.

2. There are lack of any functional assay in the in-vitro experimental. Are there any changes in the contractile function in the mouse and human iPSC cardiomyocytes carrying the mutation before and after treatment with tubastatin.

Response| We thank the reviewer for these comments. Although a vast majority of LMNA mutations result in dilated cardiomyopathy, p.H222P mutation leads to the development of Emery-Dreifuss muscular dystrophy, with skeletal muscle involvement. The C2C12 model has been widely used in the field to decipher molecular mechanisms involved in dilated cardiomyopathy caused by LMNA mutations, which have been found translatable *in vivo* later on. Readily available in the laboratory, it allowed us to perform many *in vitro* analyses in a time and cost-effective manner. We also mention at the beginning of the results section: “we utilized C2C12 cells, a simple and well-characterized model system that is frequently used as a model and can adopt some features of cardiomyocytes^{37,38}”. Despite these considerations, we fully agree with the Reviewer and we confirmed the role of tubastatin A treatment in iPSC cardiomyocytes, a model much more relevant to our problematic. We now include new Figure 5g and 5h and supplemental Figure S2 showing in iPSC-CM carrying LMNA p.H222P mutation treated with tubastatin A: 1) increased acetylated α -tubulin 2) decreased contraction frequency and 3) improved contraction profiles. We also add in the results section: “*In vitro*, treatment with tubastatin A of cardiomyocytes derived from iPSC cells (iPS-CM) from patient with p.H222P LMNA mutation was able to increase α -tubulin acetylation (Figure 5g) and decrease contraction frequency (Figure 5h) compared with untreated cells. Also, tubastatin A treatment was able to ameliorate contraction profile of iPSC-CM carrying LMNA p.H222P mutation (Figure S2), suggesting an increased contraction force. This reflects the previously reported non reciprocal relationship between force and contraction frequency in cardiac muscle cells⁵⁴⁻⁵⁶.” Given that dilated cardiomyopathy caused by LMNA mutations is characterized by high incidence of malignant tachyarrhythmias (Wahbi et al. Circulation, 2019), decreasing the contraction frequency using tubastatin A could be beneficial. This would deserve further electrophysiological studies but it seemed beyond the scope of the current paper.

3. Clinically, one of the most important clinical phenotype of LMNA cardiomyopathy is conduction defects with heart block before the occurrence of heart failure. As the authors nicely demonstrated the changes in the trafficking of Cx43 before and after tubastatin, are there any changes in ECG and occurrence of proarrhythmic phenotypes in the mouse and human iPSC model of LMNA cardiomyopathy.

Response| We thank the Reviewer for this comment. We now include a new figure and a new table (new Figure S5, new Table S9) showing information on the conduction system in the mouse models studied in our

manuscript. We now add: “Given that gap junctions are required to ensure a coordinated contraction and pumping activity in heart^{64–66}, we anticipated to prevent conduction defects (prolonged QRS complex duration) in tubastatin A-treated *Lmna*^{p.H222P/H222P} mice. One-month tubastatin A treatment failed to shorten QRS duration in 3 month-old mice (Table S9, Figure S5). Conversely, we expected to find conduction defects in *Atat1*^{-/-} mice as well as in AAV-HDAC6 transduced WT mice. Surprisingly, we did not report any conduction defects (Table S9, Figure S5). This could be explained in part by the fact that C57BL/6J genetic background mice are resistant to arrhythmia^{67–69}.”

We also add the protocol used to perform the analysis: “Electrocardiography: Electrocardiograms were recorded from mice using the non-invasive ecgTUNNEL (Emka Technologies) with minimal filtering. Waveforms were recorded using Iox Software and intervals were measured manually with ECG Auto. The electrocardiographer was blinded to mouse genotype.”

Cx43 lateralization might play a role in arrhythmias occurrence in cardiomyopathy caused by LMNA mutations, as recently suggested by our group (Le Dour *et al.* 2017; Macquart *et al.* 2019). Although *Atat1*^{-/-} and AAV-HDAC6 transduced mice did not display conduction defects, they show a proportion of Cx43 lateralization greater than in WT mice, but lower than *Lmna*^{p.H222P/H222P} mice: “To confirm the relevance of this mechanism *in vivo*, we performed immunostaining of Cx43 of cardiac sections. We observed Cx43 remodeling and redistribution to the lateral sides in *Atat1*^{-/-} mouse cardiomyocytes, similar to the phenotype observed in *Lmna*^{p.H222P/H222P} hearts (Figure 6h upper panel, 6i). Similarly, Cx43 was relocated to the lateral sides in cardiomyocytes from WT mice transduced with an AAV-HDAC6 (Figure 6h middle panel, 6i). Strikingly, Cx43 localization at intercalated discs in the hearts of *Lmna*^{p.H222P/H222P} mice was restored by treatment with tubastatin A (Figure 6h lower panel, 6i). This demonstrates that α -tubulin acetylation plays a key role in the trafficking of Cx43 to intercalated discs”. On the other hand, tubastatin A treatment restores only partially Cx43 localization but not enough to correct conduction defects in *Lmna*^{p.H222P/H222P} (Figure 6h). Lateralized localization of Cx43, leading to reduced intercellular coupling is usually associated with conduction defects. However, the relationship between electrical coupling and conduction defects is not linear, and solid lateralization of Cx43 is required for conduction defects.

Cardiac fibrosis may have additional effects to lateralized Cx43 triggering conduction defects. We now show a new Figure 6i and add in the text: “Also, abnormal tissue architecture due to increased fibrosis may have synergistic effects together with lateralization of Cx43 and lead to conduction slowing. We then performed Sirius red staining of collagen fibrils on heart sections of *Atat1*^{-/-} and *Lmna*^{p.H222P/H222P} mice treated or not with tubastatin A. As previously reported, *Lmna*^{p.H222P/H222P} mice displayed significantly increased cardiac fibrosis³⁸ as compared with WT mice (Figure 6i). Treatment with tubastatin A failed to prevent collagen fibrils accumulation (Figure 6i). On the opposite, heart sections from *Atat1*^{-/-} were similar to WT (Figure 6i), meaning that lack of acetylated α -tubulin does not trigger cardiac fibrosis.” These data corroborate the results showing that 1) tubastatin A treatment has no effect on conduction in *Lmna*^{p.H222P/H222P} mice and 2) *Atat1*^{-/-} mice have no conduction defects.

As to iPSC model, we now include new Figure 5h and new Figure S2 showing a decreased contraction frequency along with an improvement of contraction profiles in iPS-CM H222P treated with tubastatin A. *In vitro* proarrhythmic phenotype would deserve further electrophysiological studies.

4. One of the other important phenotype of LMNA cardiomyopathy is the aberrant nuclear morphology and specific disruptions in peripheral chromatin. Are there any changes in the nuclear morphology in those mutant cells as well as in the mouse cardiomyocytes?

Response| We thank the Reviewer for this comment. We now add the following: “One of the common cellular phenotypes observed in cardiomyopathy caused by mutations in LMNA is an abnormal elongation of nuclei in cardiomyocytes^{3,4,55,56}. Mean length of cardiomyocyte nuclei in hearts of *Lmna*^{p.H222P/H222P} mice treated with tubastatin A was significantly longer, as observed in untreated *Lmna*^{p.H222P/H222P} mice, than in WT mice (Figure S1a and S1b). Nuclei of cardiomyocytes in hearts of *Atat1*^{-/-} mice as well as WT mice transduced with AAV-HDAC6 had an overall shape that was “rounded”, similarly to WT mice (Figure S1a and S1b). This suggests that α -tubulin K40 acetylation does not have a role on the shape of the nuclei.” Accordingly, we add a novel Figure S1 that reports the length of cardiomyocytes nuclei in the different mouse models used in the study.

5. The authors included several human iPSC lines with LMNA mutation in this study, were those patients have both skeletal and cardiac phenotypes? Their clinical information should be included in the supplemental data.

Response| We apologize for lack of clinical information in the first version of the manuscript. This is now present in the revised version (new Table S10).

Reviewer #2 (Remarks to the Author):

In the manuscript by LeDour et al, the authors try to decipher a pathway connecting LMNA mutations in cardiomyopathy to cofilin/SRF/MRTF-A signalling and further downstream to altered tubulin acetylation and Connexin 43 mis-localization via transcriptional regulation of the ATAT1 acetyltransferase. Overall, the manuscript is well written and some of individual parts of the manuscript, particular the part on acetylation and connexin regulation are experimentally sound and very interesting (Fig. 5 and 6). However, data for several other molecular connections made in this manuscript are not always convincing. In particular, the regulation of the ATAT1 gene by SRF/MRTF which is central to their model is less robust and lacks more experimental evidence as detailed below. Furthermore, there are discrepancies between their experimental cell line C2 and the human iPSC derived cell lines from patients which suggest that their model might not have such a translational impact.

Thus, in summary some of the experimental findings are interesting and also novel and would be interesting to be published in Nature Communications. However, the individual parts are not well connected and data for some parts of their signalling model are not convincing. Thus, I would support rejection of the manuscript as it stands but encourage the authors for a major revision including several new experiments and further quantification of existing data.

Specific points:

1.) Figure 1: The data obtained in C2 cells in Figure 1 are convincing in contrast to the data presented in human iPSCs carrying a LMNA mutation (Figure 1E). Here, a cytoplasmic enrichment of MRTF-A is far from convincing and the quantification is only showing single cells. Therefore, it is unclear how relevant aberrant MRTF-A localization really is for patient-derived cells. In general, this way of quantifying immunofluorescence data (single intensity blots; Fig. 1E) is done by the authors at many positions in many Figures throughout the manuscript. This quantification does not provide any N numbers, average values, SD and statistical testing. Therefore, it is difficult to judge the robustness of data. Such a robust data analysis should be performed for all immunofluorescence data.

Response| We apologize for lack of quantification in the first version of the manuscript. We now add detailed quantification for each figure, as well as information on the statistics: Figure 1(b) “The bar graph shows the quantification of nuclear MRTF-A ($n > 200$ cells, mean \pm SD). **** $p \leq 0.0001$ ”; Figure 1(e) “The bar graph shows the quantification of nuclear MRTF-A ($n > 200$ cells, mean \pm SD). ** $p \leq 0.001$ ”; Figure 2(b) “The bar graph shows the quantification of nuclear cofilin-1 and phospho(T25)-cofilin-1 staining ($n > 200$ cells, mean \pm SD). **** $p \leq 0.0001$ ”; Figure 2(d) “The bar graph shows the quantification of nuclear MRTF-A ($n > 200$ cells, mean \pm SD). **** $p \leq 0.0001$ ”; Figure 2(f) “The bar graph shows the quantification of nuclear MRTF-A ($n > 200$ cells, mean \pm SD). **** $p \leq 0.0001$ ”.

2.) The different cofilin phosphorylation sites should be better introduced in the introduction since most people are familiar with the Ser3 site rather than the T25 site (which is very interesting). The Ser3 P site should also be analyzed in their system (Fig. 2) since one might expect reciprocal regulation to the T25 site. In Fig. 2A total cofilin levels are missing and should be provided. Fig. 2B, D, F once again lack proper quantification (see point 1). In the MRTF/cofilin IP experiment (Fig. 2h) no levels for IB: MRTF-A are provided.

Response| We thank the Reviewer for this comment. We mentioned in the discussion: “Cofilin-1 phosphorylation controls the interactions of cofilin-1 and actin and thus the mechanics of the actin regulatory machinery⁷³. The phosphorylation of cofilin-1 on Ser3 by TESK and LIMK was previously described to markedly attenuates cofilin-1/actin interactions^{74,16,75}, while dephosphorylation at this site restores the affinity of cofilin-1 for actin⁷⁶. Cofilin phosphocycling at this residue clearly drives a diverse variety of actin-dependent processes, ranging from actin-based motility⁷⁷ to the pathogenesis of neurodegenerative diseases^{78,79}”. We now better introduced Ser3 cofilin phosphorylation site by adding in introduction: “Cofilin-1 is an essential actin cytoskeleton-regulating protein¹⁵. The activity and subcellular localisation of cofilin-1 depend on its phosphorylation state. Cofilin-1 phosphorylated at serine 3 (S3) is inactive and remains in the cytosol, while the non-phosphorylated (S3) form is active and able to sever and depolymerize F-actin¹⁶”. Also, we now address

the binding between cofilin-1(T25) with a flag-tagged MRTF-A as well with the phospho(S3)-cofilin-1: “To test this hypothesis, we first transfected C2-H222P cells with Flag-tagged MRTF-A, performed immunoprecipitations (IPs) using antibodies against Flag or MRTF-A, and immunoblotted with antibody against phospho(T25)-cofilin-1. We reported an interaction between MRTF-A and phospho(T25)-cofilin-1 (Figure 2g top). Further, we found that MRTF-A interacts only with cofilin-1 and phospho(T25)-cofilin-1, but not phospho(S3)-cofilin-1, which is known to regulate cofilin-1 function⁴⁵ (Figure 2g bottom).” As suggested by the reviewer, we added Ser3 phospho site analysis in Figure 2a and 2g and total cofilin levels in Figure 2a. Proper quantifications were added for Figure 2b, d and f. MRTF-A immunoblots for immunoprecipitation experiments were provided in Figure 2g and 2h.

3.) My major concern is with the regulation of ATAT1 gene by SRF/MRTF-A (Fig. 4). Here, the authors only provide luciferase assays which are highly artificial and are in my point of view not sufficient to show a clear regulation of ATAT1 by SRF/MRTF-A that can support convincingly the author’s model. Also, the authors refer to a previous paper by Fernandez-Barrera (JCB) for the ATAT1 regulation by SRF. But also in this paper only luciferase assays were performed. The authors should use SRF/MRTF loss- (mutant cells or siRNA as done for cofilin phosphatases) and gain-of-function(e.g. SRF-VP16 overexpression) to show changes in ATAT1 endogenous mRNA regulation. Also, ChIP should be performed in a relevant cell line to clearly document ATAT1 promoter occupancy by SRF

Response| We thank the Reviewer for this comment. We now show regulation of *ATAT1* expression by loss-of-function (MRTFΔ100) and gain of function (SRF-VP16 and MRTF) transfections *in vitro*. We reported these results in new Figure 4d and added in the results section: “To demonstrate causal relationship of MRTF-A/SRF on *ATAT1* transcription, we performed gain and loss-of-function experiments. For gain of function, we transfected C2C12 cells with plasmid expressing constitutively active SRF-VP16 (a fusion protein of SRF and the viral VP16 transactivation domain) or with WT MRTF-A. For loss of function, cells were transfected with MRTFΔ100, an MRTF-A deleted in a region required for its nuclear import and transcriptional activity. SRF-VP16 and MRTF transfections increased *Atat1* along with *Srf* and *Acta2* expression. Concordantly, transfection with MRTFΔ100 had no effect on these three genes expression. These results confirm that MRTF-A/SRF axis regulates *Atat1* gene expression (Figure 4d).”

Unfortunately, we were not able to succeed in ChIP experiments in the allocated time for the revision of our manuscript. We couldn’t find literature-confirming SRF binding to *Atat1* promoter (https://maayanlab.cloud/Harmonizome/gene_set/SRF/ENCODE+Transcription+Factor+Targets). However, unlike the ChIP or EMSA assays, which only assess the ability of a protein to interact with a region of DNA, a luciferase assay is able to establish a functional connection between the presence of the protein and the amount of gene product that is produced. Our dataset generated a beam of evidence showing MRTF-A/SRF axis is involved, if not directly, at least indirectly in *Atat1* regulation and subsequent α-tubulin acetylation. The precise regulation mechanism would require further studies.

In Fig. 4F there are 5 samples for LMNA patients in the blot, however only four are quantified. How come this discrepancy?

Response| This issue is now fixed in the revised manuscript.

Minor point: on page 7 line 205 the authors refer to previous findings but it is not obvious what they are relating to (Figure, paper?).

Response| We thank the Reviewer for this comment. This is now fixed in the revised manuscript and we removed the following sentence: “Consistent with previous findings, this indicates that inhibiting ERK1/2 phosphorylation prevents decreased SRF activity.”

Reviewer #3 (Remarks to the Author):

The manuscript by Le Dour et al. describes the cellular consequences of mutations in the lamin A/C gene which is a cause of dilated cardiomyopathy. Both effects on the actin cytoskeleton as well as on the microtubules were already observed before and here both observations are connected. Phosphorylated cofilin 1 (by ERK1/2) binds to myocardin-related transcription factor 1 (MRTF-A) thus preventing the stimulation of serum response factor

(SRF) in the nucleus. This leads to the decreased alpha-tubulin acetylation by a reduced expression of alpha-tubulin acetyltransferase A. Increasing alpha tubulin acetylation by inhibiting HDAC6 has an interesting therapeutic effect.

Major remarks

- While after reading this manuscript the impression exists that the observation of defects in acetylation of microtubules and the connection with the mislocalization of Cx43 are a completely new observation, a (partially) similar conclusion was formulated in the 2019 HMG paper of the same research group (“...microtubule cytoskeleton alteration and decreased acetylation of α -tubulin lead to remodeling of Cx43 in LMNA cardiomyopathy, which alters the correct communication between cardiomyocytes,...). This paper is mainly cited in the current manuscript to make the point that mutations in LMNA are associated with microtubule network alterations. In fact, much more evidence is present in the HMG manuscript linking already acetylation of microtubules with the observed phenotypes. Moreover, the potential therapeutic role of paclitaxel is not mentioned in the current manuscript. As a consequence, a much better integration/discussion of the results obtained and published before is crucial.

Response| We agree with the reviewer that our study in Macquart et al., Hum Mol Genet 2019 already mentioned microtubule instability (evidenced by decreased acetylation levels) and Cx43 mislocalization in cardiomyopathy caused by mutations in LMNA. However, we now focused on this peculiar PTM (rather than being a hallmark of microtubule instability). We evidenced the mechanism involved in decreased acetylation of microtubules in cardiomyopathy caused by mutations in LMNA using a combination of cell and animal models including iPS-CM. In addition, we studied cardiac phenotype of *Atat*^{-/-} mouse model, linking for the very first time a lack of acetylated microtubules to cardiac dysfunction. Also, we now show heart malfunction in WT mice transduced with AAV overexpressing HDAC6 and add in the text: “Given that the deacetylation of α -tubulin is driven by HDAC6^{51,52}, and to confirm the role of α -tubulin K40 acetylation on cardiac function, we studied by echocardiography the cardiac function of WT mice transduced with or without an AAV vector expressing HDAC6 (AAV-HDAC6). Compared to WT mice, mice transduced with AAV-HDAC6 had decreased acetylation level of α -tubulin (Figure 5c). This is ensuing a significantly increased left ventricular end-systolic diameters and decreased fractional shortening (Table 2, Figure 5d). Taken together, these data highlight the importance of α -tubulin K40 acetylation for cardiac function.” Finally, based on the data we generated, we now used tubastatin A treatment which is specifically targeting microtubule acetylation through HDAC6 inhibition rather than paclitaxel stabilizing microtubules by binding β -tubulin subunits and interacting with other microtubule associated proteins. We do feel we bring much more evidence in the present manuscript linking decreased acetylation of microtubules to cardiac dysfunction.

Still, we now discuss in greater details the potential use of microtubule-stabilizing agents as therapeutic approach: “Our recent work showed that increasing α -tubulin K40 acetylation by augmenting microtubule stability using paclitaxel, could also be potentially translatable into therapy for cardiomyopathy caused by mutations in LMNA⁹. It has been showed that paclitaxel can be safely administered in patients with cardiac dysfunction⁹⁹. Other microtubule-stabilizing agents have been described and should also benefit clinical development for cardiac diseases¹⁰⁰.”

- While the title focuses on the actin-microtubule interplay, the consistently lower alpha-tubulin acetylation (published already before in the HMG paper) but mainly the therapeutic potential of modulating this tubulin acetylation by inhibiting HDAC6 deserves much more attention. At present, this therapeutic effect is only mentioned at the end of the abstract without even linking it to HDAC6.

Response| We thank the Reviewer for this comment. We now introduced the therapeutic potential of modulating this tubulin acetylation by inhibiting HDAC6. We add: “Given that the deacetylation of α -tubulin is driven by HDAC6^{52,53}, and to confirm the role of α -tubulin K40 acetylation on cardiac function, we studied by echocardiography the cardiac function of WT mice transduced with or without an AAV vector expressing HDAC6 (AAV-HDAC6). Compared to WT mice, mice transduced with AAV-HDAC6 had decreased acetylation level of α -tubulin (Figure 5c). This is ensuing a significantly increased left ventricular end-systolic diameters and decreased fractional shortening (Table 2, Figure 5d). Taken together, these data highlight the importance of α -tubulin K40 acetylation for cardiac function.” We report in a new Table 2 the echocardiographic data for WT mice transduced or not with AAV expressing HDAC6.

Moreover, it seems to be crucial to find out how the reduction in alpha-tubulin acetylation is mechanistically linked to a loss of Cx43 localization at intercalated discs.

Response| We thank the Reviewer for this comment. To further decipher the role of altered microtubules on Cx43 localization, we now add a new figure (FigureS4). Given that acetylated tubulin has been shown to be necessary for trafficking of molecules along the microtubules, we sought to demonstrate that in LMNA cardiomyopathy, the trafficking of Cx43 is dependent of microtubules: “*The trafficking of Cx43 along microtubules is driven in part by motor proteins from the kinesin-1 family*⁶⁰. *It has also been shown that α -tubulin acetylation positively influences motor protein binding to microtubules and has a role in the intracellular trafficking*⁶¹. *To determine if active, microtubule-based transport is important for Cx43 localization, we treated cells with adenosine 5'-(β,γ -imido)triphosphate (AMP-PNP), a non-hydrolysable analogue of ATP that inhibits motility of kinesin motor proteins that drive intracellular transport. Following AMP-PNP treatment, Cx43 was misplaced within the cytoplasm of C2-WT myoblasts (Figure S4a) and cardiomyocytes isolated from WT mice (Figure S4b). To test the role of kinesin in this process more directly, we transfected C2-WT myoblasts with siRNA against Kif5B, the kinesin motor suggested to be responsible for Cx43 trafficking (Figure S4c). In cells depleted of Kif5B, Cx43 was localized in the cytosol rather than at cell-cell junctions (Figure S4d). This demonstrates an important role of active microtubule-based transport in the trafficking and localization of Cx43 to cell-cell junctions. This process is perturbed in the presence of mutated A-type lamins”*

Minor remarks

- While it is clearly shown that tubastatin A has a positive effect on acetylation of alpha-tubulin, which subsequently also has a positive effect on the Cx43 localization, it needs to be excluded that at least part of this effect is due to a non-specific effect of tubastatin A on other HDACs. This can be easily proven by determining the (absence of an) effect of tubastatin A on histone acetylation.

Response| We totally agree and we thank the reviewer for this comment. We now show mRNA expression levels of other HDACs in hearts of *Lmna*^{H222P/H222P} mice treated or not with tubastatin A. We now add a new figure (Figure S1) and we add in the text: “*Tubastatin A treatment significantly decreased expression of Hdac6 without altering other Hdacs expression (Figure S1) and increased the level of α -tubulin acetylation in hearts of *Lmna*^{p.H222P/H222P} mice as compared to DMSO treated, as shown by the immunoblotting of cardiac tissue homogenates (Figure 5e)*”.

- Tubastatin A (not tubastatin) should be introduced much better in the abstract.

Response| We thank the reviewer for this comment. This is now done in the new manuscript: “*Furthermore, increasing tubulin acetylation levels in *Lmna*^{p.H222P/H222P} mice with tubastatin A treatment to inhibit the HDAC6 activity, restored the proper localization of Cx43 and improved cardiac function*”. Also, we replaced tubastatin with tubastatin A.

- The acetylation level of alpha tubulin has many more effects on microtubule than just stabilizing them. This should be discussed in more detail.

Response| We thank the Reviewer for this comment. We now add a new paragraph on the effects of tubulin acetylation: “*Post-translational modifications can play important roles in controlling the stability and function of microtubule. Most of tubulin post-translational modifications (e.g., detyrosination, polyglutamylolation, and polyglycylation) alter residues within the C-terminal tail of tubulin that extends from the surface of the microtubule. However, α -tubulin K40 acetylation is the unique tubulin post-translational modification that localizes to the inside of the microtubule. Acetylation of α -tubulin K40 was found to stable, long-lived microtubule subpopulations*^{31,90}. *However, it still remains controversial whether the relationship between α -tubulin K40 acetylation and microtubule stabilization is correlative or causative.*”

- Is the expression level of HDAC6 affected by the presence of LMNA mutations?

Response| Given that HDAC6 is the main deacetylase for lysine 40 on tubulin, it is crucial to address the expression level of this HDAC in biological samples expressing *LMNA* mutations, as suggested by the reviewer. We now add a new Figure 4k describing the lack of HDAC6 expression alterations in the cells and cardiac tissues used in our study: “*Histone deacetylase 6 (HDAC6) is the major deacetylase responsible for removing the acetyl group of α -tubulin*^{50,51}. We did not report any modulation in HDAC6 expression neither in the *LMNA*-mutated heart tissue compared with control heart tissue nor in C2-H222P cells compared with C2-WT cells (Figure 4k).”

- A broader discussion on the therapeutic possibilities and limitations of HDAC6 inhibition should be included in the discussion. At present, only a reference to a review on HDAC6 in cancer is included. This is insufficient, especially given the fact that tubastatin A is not used clinically to treat cancer. Moreover, tubastatin A belongs to a chemical class of molecules which could compromise its clinical use.

Response| We now further discuss the potential therapeutic role of HDAC inhibitors in *LMNA* cardiomyopathy: “*Aberrant tubulin acetylation profiles are observed in a broad spectrum of diseases*⁹⁵, which has led to using HDAC inhibition as a therapeutic approach⁹⁶. Several reports have described beneficial effects of HDAC inhibition in the progression of neurodegenerative phenotypes⁹⁷. The therapeutic potential of HDAC inhibition has also been explored for different cancers and the HDAC inhibitor, suberoylanilide hydroxamic acid (SAHA), is FDA-approved to treat cutaneous T-cell lymphoma. However, SAHA is a pan-HDAC inhibitor that has activity toward several classes of HDACs. Therefore, there is a need to develop specific HDAC inhibitors. Tubastatin A is not optimized for oral delivery and therefore will not be suitable for clinical trials. Among HDAC6 inhibitors, only ricolinostat (ACY-1215, ricolinostat) is currently evaluated in clinical trials for several forms of cancer^{36,98}. Given that HDAC6 inhibitors have lower adverse effects than pan-HDAC inhibitors, such compounds may be important in the array of therapeutic approaches for cardiomyopathy.”

We would like to thank the experts again for taking the time to review our manuscript. We hope the substantial revision of our work will convince the reviewers that our manuscript is suitable for a publication in *Nature Communications*.

REVIEWER COMMENTS

Reviewer #1 (Remarks to the Author):

The revised paper has addressed my concerns

Reviewer #2 (Remarks to the Author):

In the revised manuscript, the authors addressed several concerns sufficiently particularly those raised regarding Figure 1 and 4.

However, one major issue remains to me in regard to Figure 2A. Here the authors now provide total cofilin levels and P-Ser3 levels. Looking carefully at the immunoblots, total cofilin levels are clearly raised in C2-H222P compared to WT. In fact, effects appear to be even stronger as for the P-T25 cofilin. In contrast p-cofilin- S3 appears unaltered. Thus, when normalizing P-cofilin levels to total cofilin for quantification I am somewhat surprised that T25 phosphorylation still goes up. Given that total cofilin is also upregulated I would hardly expect any change for T25 phosphorylation. Having said this, p-cofilin S3 levels could actually be decreased in C2-H222P compared to WT since overall P-Cofilin S3 is unaltered but overall cofilin levels go up in C2-H222P cells. This issue should be resolved and commented on by the authors

Reviewer #3 (Remarks to the Author):

The revision of the manuscript of Ledour et al. solved a number of problems. However, some of my remarks were not properly considered. More importantly, rewriting the manuscript and incorporating new data induced some important confusion which should be solved.

- Some (important) confusion is introduced by adding additional references. While asking for a broader discussion on the potential therapeutic role of HDAC6 in disease, references are included describing the therapeutic effect of other HDAC inhibitors (4b and SAHA) in Huntington disease. These inhibitors indeed modify the histone acetylation which are not due to HDAC6 (as this is a cytoplasmic enzyme). The inhibitors used in the context of these studies affect the expression pattern of many different targets as these are not selective inhibitors of HDAC6. This could confuse the readers and should be adapted.

- The extra data on the expression level of HDAC6 after a treatment with tubastatin A are puzzling (Fig. S1). It is not expected that HDAC6 inhibition alters the expression level of HDAC6 unless this inhibition has non-specific effects on other targets (e.g. histones). That was exactly the reason why I asked to determine the (absence of an) effect of tubastatin A on histone acetylation. These experiments should be done and an explanation for the decrease in the mRNA expression of HDAC6 after treatment with tubastatin A should be given as this is not what one expects from an inhibitor of the deacetylating function of HDAC6. As far as I am aware this was never shown before. Moreover, this effect should also be confirmed at the protein level.

- My question to discuss the problems related to the fact that tubastatin A belongs to a class of molecules (hydroxamates) which can't be used in patients because of their potential genotoxicity is not addressed in the best way possible. This is important as it has major implications to translate the findings described in this manuscript to a potential therapeutic strategy.

- In relation to my (minor) remark that tubastatin A should be introduced in a better way, the least one could expect is a reference to the paper describing for the first time the synthesis and characterization of this compound (Butler et al., PMID: 20614936).

- "Aberrant tubulin acetylation profiles are observed in a broad spectrum of diseases" is followed by a reference related to cardiac proteotoxicity. The spectrum of diseases is much broader as well as the

potential use of HDAC6 inhibition in several of these diseases. In addition, the proposed mechanism of action described in the cited paper isn't the same as the one described in the current manuscript and this should also be discussed in more detail.

- The numbers of the references in the 'answers to the referees' is different than in the revised manuscript which doesn't make it always easy to find out which extra references were added.

Actin-microtubule cytoskeletal interplay mediated by MRTF-A/SRF signaling promotes dilated cardiomyopathy caused by LMNA mutation

Response to reviewers' comments

We thank again the reviewers for their valuable time and suggestions to review our paper. We have addressed the reviewers' concerns and incorporated their new suggestions in the revised version. A point-by-point response to the reviewers' comments follows.

Reviewer#1 (Remarks to the Author):

The revised paper has addressed my concerns

Response| We thank Reviewer#1.

Reviewer#2 (Remarks to the Author):

In the revised manuscript, the authors addressed several concerns sufficiently particularly those raised regarding Figure 1 and 4.

1. However, one major issue remains to me in regard to Figure 2A. Here the authors now provide total cofilin levels and P-Ser3 levels. Looking carefully at the immunoblots, total cofilin levels are clearly raised in C2-H222P compared to WT. In fact, effects appear to be even stronger as for the P-T25 cofilin. In contrast p-cofilin- S3 appears unaltered. Thus, when normalizing P-cofilin levels to total cofilin for quantification I am somewhat surprised that T25 phosphorylation still goes up. Given that total cofilin is also upregulated I would hardly expect any change for T25 phosphorylation. Having said this, p-cofilin S3 levels could actually be decreased in C2-H222P compared to WT since overall P-Cofilin S3 is unaltered but overall cofilin levels go up in C2-H222P cells. This issue should be resolved and commented on by the authors.

Response| We thank Reviewer#2 for this comment. In a recent work, we report increased cofilin-1 expression in LMNA mutant muscle cells. We demonstrated that hyperactivated ERK1/2 signaling prevents cofilin-1 degradation through the proteasome⁴³ (doi: 10.1016/j.celrep.2021.109601). Hence, the data presented in Figure 2A of this current work are in line with published results. We agree with the reviewer that increased expression of the total form of cofilin 1 can be puzzling to evaluate an increase of phosphorylated levels. However, while quantifications show a 3-fold increase for total cofilin 1, it shows a 4.5-fold increase for P (Thr25) cofilin, meaning that even if total cofilin 1 expression is elevated, its relative phosphorylation is increased as well. We now add in the text: “The observed overexpression of total cofilin-1 in C2-H222P cells (Figure 2a) results from impaired degradation by the proteasome as previously described⁴³ (doi: 10.1016/j.celrep.2021.109601)”. We did not detect changes in (Ser3)cofilin-1 expression. This is in accordance with Chatzifrangkeskou et al. 2018 (doi: 10.1093/hmg/ddy215), in which analysis by mass spectrometry showed similar phosphorylated (Ser3) cofilin-1 expression level in cells expressing either wild type or mutant lamin A.

Reviewer#3 (Remarks to the Author):

The revision of the manuscript of Le Dour et al. solved a number of problems. However, some of my remarks were not properly considered. More importantly, rewriting the manuscript and incorporating new data induced some important confusion which should be solved.

1. Some (important) confusion is introduced by adding additional references. While asking for a broader discussion on the potential therapeutic role of HDAC6 in disease, references are included describing the therapeutic effect of other HDAC inhibitors (4b and SAHA) in Huntington disease. These inhibitors indeed modify the histone acetylation which are not due to HDAC6 (as this is a cytoplasmic enzyme). The inhibitors used in the context of these studies affect the expression pattern of many different targets as these are not selective inhibitors of HDAC6. This could confuse the readers and should be adapted.

Response| We thank Reviewer#3 for this comment. Indeed, we agree that the additional references could raise confusion in readers mind, even though we clearly mentioned that “SAHA is a pan-HDAC inhibitor that has

activity toward several classes of HDACs”, we only wanted to point out “there is a need to develop specific HDAC inhibitors”. We removed the following confusing part: “Several reports have described beneficial effects of HDAC inhibition in the progression of neurodegenerative phenotypes. The therapeutic potential of HDAC inhibition has also been explored for different cancers and the HDAC inhibitor, suberoylanilide hydroxamic acid (SAHA), is FDA-approved to treat cutaneous T-cell lymphoma. However, SAHA is a pan-HDAC inhibitor that has activity toward several classes of HDACs. Therefore, there is a need to develop specific HDAC inhibitors.” We now focused the discussion on HDAC6, modified the text accordingly and cited new references: “Aberrant tubulin acetylation profiles are observed in a broad spectrum of diseases^{94,98,100} (doi: 10.1007/s00795-020-00260-8; doi: 10.1007/s00018-015-2000-5; doi: 10.1016/j.cub.2017.10.044), which has led to using HDAC6 inhibition as a therapeutic approach. General HDAC inhibitors lead to the deacetylation of histones, modify gene transcription, and increase the acetylation of many non-histone proteins^{101,102} (doi: 10.1016/j.tips.2010.09.003; doi: 10.2174/1568026615666150825125857). HDAC6 does not act on histones and its major substrate is α -tubulin¹⁰³ (doi: 10.1038/sj.onc.1210614). Acetylated α -tubulin occurs on polymerized microtubules and affects microtubules dynamics and stability³¹ (doi: 10.1038/s41580-020-0214-3). Hence, the finding that α -tubulin acetylation in the heart is involved in cardiomyopathy, and that specific inhibition of HDAC6 has beneficial effects, opens up new therapeutic possibilities.”

2. The extra data on the expression level of HDAC6 after a treatment with tubastatin A are puzzling (Fig. S1). It is not expected that HDAC6 inhibition alters the expression level of HDAC6 unless this inhibition has non-specific effects on other targets (e.g. histones). That was exactly the reason why I asked to determine the (absence of an) effect of tubastatin A on histone acetylation. These experiments should be done and an explanation for the decrease in the mRNA expression of HDAC6 after treatment with tubastatin A should be given as this is not what one expects from an inhibitor of the deacetylating function of HDAC6. As far as I am aware this was never shown before. Moreover, this effect should also be confirmed at the protein level.

Response| We thank Reviewer#3 for this comment. Decreased *Hdac6* mRNA expression in heart samples of mice treated with tubastatin A appears puzzling. However this was not found at the protein level. Since α -tubulin is a specific substrate for HDAC6 but not other HDACs, the selective inhibition of HDAC6 can be characterized by hyperacetylation of tubulin (shown in Figure 5e) without affecting the level of histone acetylation. We now replace quantification of mRNA expression in Supplemental Figure 1 by western blot analysis and corresponding quantifications showing: 1) unchanged HDAC6 protein expression, 2) unchanged expression of two other nuclear HDACs: HDAC3 and HDAC4, 3) we confirmed that tubastatin A treatment did not affect acetylation levels of histone H3 (AcH3K9 and AcH3K27) as previously described (Butler et al., 2010, doi: 10.1021/ja102758v). This confirms tubastatin A treatment acts without non-specific effects on nuclear HDAC or histone acetylation. We now change in the text: “Tubastatin A treatment increased the level of alpha-tubulin acetylation in hearts of *Lmna*^{p.H222P/H222P} mice as compared with DMSO treatment (Figure 5e) without altering other HDAC expression or acetylation of histone H3 (Figure S1).” We also add this new paragraph that opens up scientific perspectives of our work: “While α -tubulin is the main target of HDAC6, this latter has other substrates including cortactin and ERK1 that has been shown to be dysregulated in LMNA-cardiomyopathy^{104,105} (doi: 10.1016/j.phrs.2020.105274; doi: 10.1016/j.bbrc.2020.05.102). It has also been reported that HDAC6 inhibition has a cardioprotective effect through autophagy induction, a mechanism also affected in LMNA-cardiomyopathy^{98,106} (doi: 10.1073/pnas.1415589111; doi: 10.1073/pnas.1415589111). A synergistic effect through α -tubulin, cortactin, ERK1 and autophagy induction might be highly beneficial in LMNA-cardiomyopathy.” This raise interesting scientific hypotheses, which are out of the scope of the current work but should be further investigated.

3. My question to discuss the problems related to the fact that tubastatin A belongs to a class of molecules (hydroxamates) which can't be used in patients because of their potential genotoxicity is not addressed in the best way possible. This is important as it has major implications to translate the findings described in this manuscript to a potential therapeutic strategy.

Response| We thank Reviewer#3 for this comment. We agree with Reviewer#3, this is an important point. In the present work, we used tubastatin A more as a tool to restore level of acetylated α -tubulin, than as a therapeutic option. This could be the main focus of a following study. We modified the text accordingly and hope we addressed it in a better way: “Most of the promising HDAC inhibitors target multiple HDAC isoforms. These pan-HDAC inhibitors are often driving several undesirable side effects, which limits the clinical utility of these molecules. Selective HDAC6 inhibitor tubastatin A⁵⁵ (doi.org/10.1021/ja102758v) may exhibit reduced side effects compared to the pan-HDAC inhibitors. However tubastatin A belongs to hydroxamate class of molecules and display mutagenicity/genotoxicity and poor oral delivery, thus hindering its development for clinical use¹⁰⁷

(doi: 10.1002/cmdc.201500486). Therefore, the development of a highly potent and selective HDAC6 inhibitor with good bioavailability and without genotoxicity is highly desired. Access to such HDAC6 inhibitor would enable us to study the effects of HDAC6 inhibition in cardiomyopathy caused by mutations in LMNA in preclinical in vivo models for translation to clinical trials. Recent researches led to the identification of novel HDAC6 inhibitors that could drive further mechanistic exploration and definitive clinical translation in cardiomyopathy^{108,109,110} (doi: 10.1073/pnas.1313893110; doi: 10.1073/pnas.1515882112; doi: 10.1021/cb200134p).”

4. In relation to my (minor) remark that tubastatin A should be introduced in a better way, the least one could expect is a reference to the paper describing for the first time the synthesis and characterization of this compound (Butler et al., PMID: 20614936).

Response| We thank Reviewer#3 for this comment. We modified this accordingly and now cite the seminal paper: “We next assessed whether increasing the acetylation levels in *Lmna*^{p.H222P/H222P} mice with the HDAC6 inhibitor tubastatin A⁵⁵ (doi.org/10.1021/ja102758v) could improve the cardiac function in these animals.”

5. “Aberrant tubulin acetylation profiles are observed in a broad spectrum of diseases” is followed by a reference related to cardiac proteotoxicity. The spectrum of diseases is much broader as well as the potential use of HDAC6 inhibition in several of these diseases. In addition, the proposed mechanism of action described in the cited paper isn’t the same as the one described in the current manuscript and this should also be discussed in more detail.

Response| We thank Reviewer#3 for this comment. Again, in the present work, we used HDAC6 inhibitor, tubastatin A, more as a tool to restore level of acetylated α -tubulin, than as a therapeutic option *per se*. We now modified the text accordingly and cite new references: “Aberrant tubulin acetylation profiles are observed in a broad spectrum of diseases^{94,98-100} (doi: 10.1016/j.cub.2017.10.044; doi: 10.1073/pnas.1415589111; doi: 10.1007/s00795-020-00260-8; doi: 10.1007/s00018-015-2000-5), which has led to using HDAC inhibition as a therapeutic approach. General HDAC inhibitors lead to the deacetylation of histones and thus influence gene transcription and they increase the acetylation of many non-histone proteins^{101,102} (doi: 10.1038/nrd268; doi: 10.1016/j.tips.2010.09.003). HDAC6 does not act on histones and its major substrate is α -tubulin¹⁰³ (doi: 10.1038/sj.onc.1210614). Acetylated α -tubulin occurs on polymerized microtubules and affects microtubules dynamics and stability³¹ (doi.org/10.1038/nrm3227). Hence, the finding that HDAC6 in the heart is involved in cardiomyopathy, and that its specific inhibition has beneficial effects, opens up new therapeutic possibilities using isoform-specific inhibitors.” Although reference⁹⁸ describes a different mechanism of action with a genetic cardiomyopathy associated with hyperacetylated tubulin, HDAC6 inhibition still improved cardiac function in a mouse model of the disease. To further discuss, we now add in the text: “While α -tubulin is the main target of HDAC6, this latter has other substrates including cortactin and ERK1 that has been shown to be dysregulated in LMNA-cardiomyopathy^{104,105} (doi: 10.1016/j.phrs.2020.105274; doi: 10.1016/j.bbrc.2020.05.102). It has also been reported that HDAC6 inhibition has a cardioprotective effect through autophagy induction, a mechanism also affected in LMNA-cardiomyopathy^{98,106} (doi: 10.1073/pnas.1415589111; doi: 10.1073/pnas.1415589111). A synergistic effect through α -tubulin, cortactin, ERK1 and autophagy induction might be highly beneficial in LMNA-cardiomyopathy.”

6. The numbers of the references in the ‘answers to the referees’ is different than in the revised manuscript which doesn’t make it always easy to find out which extra references were added.

Response| We thank Reviewer#3 for this comment. We are sorry for the inconvenience and we addressed it by putting doi in addition to the references numbers.

We would like to thank the experts again for taking the time to review our manuscript. We hope the revision of our work will convince the reviewers and that our manuscript is suitable for a publication in *Nature Communications*.

REVIEWERS' COMMENTS

Reviewer #2 (Remarks to the Author):

The authors tried to clarify the remaining questions but only succeeded in part. I am not questioning the up-regulation of total cofilin levels but my point was that this up-regulation comes with consequences for P-cofilin levels and the interpretation thereof. Thus it might be that P-Cofilin T25 is still stronger up but at the same time P-S3 cofilin levels might be decreased. This has not been sufficiently addressed and I am not sure whether the comment now included by the authors will help readers stumbling across the quantification issue raised for Fig. 2A.

Reviewer #3 (Remarks to the Author):

The authors have done many efforts to clarify the confusion that was introduced during the first revision. Especially, the new supplementary figure 1 is very helpful.

I only have two (important but minor) remarks remaining.

In the rewritten part of the discussion, the statement that 'Aberrant tubulin acetylation profiles'...'has led to using HDAC6 inhibition as a therapeutic approach' (line 524) should be followed by references. Does this refer to the clinical trials in cancer using HDAC6 inhibitors? In that case, the therapeutic effect isn't linked to correcting tubulin acetylation (as far as I know). Alternatively, this statement could refer to (mouse?) studies in other diseases. This important statement should be clarified by adding extra information.

Apart from the typo in the statement that 'Recent researches led to the identification of novel HDAC6 inhibitors that could drive further mechanistic exploration and definitive clinical translation in cardiomyopathy, most of the references following this (new) statement are from almost 10 years ago. In addition, at least two of these studies refer to compounds belonging to the same chemical class as tubastatin A (hydroxamates). Not clear whether the third study provides a solution as this compound also inhibits NFκB and inhibits HDAC6 only in the μM range. I would suggest removing this statement.

Actin-microtubule cytoskeletal interplay mediated by MRTF-A/SRF signaling promotes dilated cardiomyopathy caused by LMNA mutation

Response to reviewers' comments

We thank again the reviewers for their comments on our paper. We have addressed the reviewers' last concerns and incorporated their new suggestions in the revised version. A point-by-point response to the reviewers' comments follows.

REVIEWERS' COMMENTS

Reviewer #2 (Remarks to the Author):

The authors tried to clarify the remaining questions but only succeeded in part. I am not questioning the up-regulation of total cofilin levels but my point was that this up-regulation comes with consequences for P-cofilin levels and the interpretation thereof. Thus it might be that P-Cofilin T25 is still stronger up but at the same time P-S3 cofilin levels might be decreased. This has not been sufficiently addressed and I am not sure whether the comment now included by the authors will help readers stumbling across the quantification issue raised for Fig. 2A.

We agree with Reviewer #2 and hope we now clarify this point. We now add in the text: "Although we show overexpression of total cofilin-1 in C2-H222P cells (Figure 2a) (previously described as resulting from impaired degradation by the proteasome (*Vignier et al., Cell Rep, 2021*)), expression of cofilin-1 phosphorylated on S3 was unchanged, suggesting its relative decreased phosphorylation. In the meantime, we confirmed that the phosphorylation of cofilin-1 on T25 was significantly increased in C2-H222P cells, as previously reported in *Chatzifrangkeskou et al., Hum Mol Genet, 2018* (Figure 2a)."

Reviewer #3 (Remarks to the Author):

The authors have done many efforts to clarify the confusion that was introduced during the first revision. Especially, the new supplementary figure 1 is very helpful. I only have two (important but minor) remarks remaining.

In the rewritten part of the discussion, the statement that 'Aberrant tubulin acetylation profiles'... 'has led to using HDAC6 inhibition as a therapeutic approach' (line 524) should be followed by references. Does this refer to the clinical trials in cancer using HDAC6 inhibitors? In that case, the therapeutic effect isn't linked to correcting tubulin acetylation (as far as I know). Alternatively, this statement could refer to (mouse?) studies in other diseases. This important statement should be clarified by adding extra information.

We fully agree with Reviewer #3. We now change the text and add references: "Aberrant tubulin acetylation profiles are observed in a broad spectrum of diseases^{94,98-100}, which has led to using HDAC6 inhibition as a therapeutic approach in preclinical studies¹⁰¹⁻¹⁰⁵ (10.1002/adbi.202101308 ; 10.1038/s41598-021-94923-w ; 10.7554/eLife.63076 ; 10.1016/j.ajpath.2020.08.013 ; 10.1016/j.expneurol.2020.113281)".

Also, a recent paper from Osseni et al. (<https://doi.org/10.1038/s41467-022-34831-3>) sheds light on the use of HDAC6 inhibitor tubastatin A in a mouse model of neuromuscular disorder through Smad3 acetylation. Given that TGF β /Smad2/3 pathway is also affected in our mouse model of *LMNA*-cardiomyopathy, we now add in the text: "Furthermore, while α -tubulin is the main target of HDAC6, this latter has other substrates including cortactin, ERK1 and Smad3 that have been shown to be dysregulated in *LMNA*-cardiomyopathy^{39,109-112},"

Apart from the typo in the statement that 'Recent researches led to the identification of novel HDAC6 inhibitors that could drive further mechanistic exploration and definitive clinical translation in cardiomyopathy, most of the references following this (new) statement are from almost 10 years ago. In addition, at least two of these studies refer to compounds belonging to the same chemical class as tubastatin A (hydroxamates). Not clear whether the third study provides a solution as this compound also inhibits NF κ B and inhibits HDAC6 only in the μ M range. I would suggest removing this statement.

We agree with Reviewer #3 and now remove the following sentence: "Recent researches led to the identification of novel HDAC6 inhibitors that could drive further mechanistic exploration and definitive clinical translation in cardiomyopathy." and associated references.